# Assessing Impact Factors That Affect School Mobility Utilizing a Machine Learning Approach

Stylianos Kolidakis [1,*] , Kornilia Maria Kotoula [2] , George Botzoris [3] , Petros Fotios Kamberi [1] and Dimitrios Skoutas [1]

1   Athena Research and Innovation Center, Information Management Systems Institute, Artemidos 6 and Epidavrou, 15125 Marousi, Greece; pkamberi@athenarc.gr (P.F.K.); dskoutas@athenarc.gr (D.S.)
2   Centre for Research and Technology Hellas, Hellenic Institute of Transport, 6th Km Harilaou–Thermi, 57001 Thessaloniki, Greece; nilia@certh.gr
3   Department of Civil Engineering, Section of Transportation, Kimmeria Campus, Democritus University of Thrace, 67100 Xanthi, Greece; gbotzori@civil.duth.gr
*   Correspondence: skolidakis@athenarc.gr

**Abstract:** The analysis and modeling of parameters influencing parents' decisions regarding school travel mode choice have perennially been a subject of interest. Concurrently, the evolution of artificial intelligence (AI) can effectively contribute to generating reliable predictions across various topics. This paper begins with a comprehensive literature review on classical models for predicting school travel mode choice, as well as the diverse applications of AI methods, with a particular focus on transportation. Building upon a published questionnaire survey in the city of Thessaloniki (Greece) and the conducted analysis and exploration of factors shaping the parental framework for school travel mode choice, this study takes a step further: the authors evaluate and propose a machine learning (ML) classification model, utilizing the pre-recorded parental perceptions, beliefs, and attitudes as inputs to predict the choice between motorized or non-motorized school travel. The impact of potential changes in the input values of the ML classification model is also assessed. Therefore, the enhancement of the sense of safety and security in the school route, the adoption of a more active lifestyle by parents, the widening of acceptance of public transportation, etc., are simulated and the impact on the parental choice ratio between non-motorized and motorized school commuting is quantified.

**Keywords:** machine learning; artificial intelligence; mode choice forecast; school transportation; sustainable mobility; school travel mode choice modeling

## 1. Introduction

### 1.1. Rationale

School travel mode choice modeling plays a pivotal role in a city's transportation planning procedure, as students' travel activities are a significant and crucial part of everyday life. Its main objective focuses on forecasting transport mode preferences for students of various characteristics and also parents, who in most cases are responsible for the student's transport mode choice [1]. In general, an accurate travel mode prediction benefits the traffic demand prediction and the market share ratio prediction, as well as the associated traffic congestion alleviation [2]. This undoubtedly applies to the school trips completion forecasting process as well.

Additionally, school travel mode choice modeling is an imperative need for shaping the appropriate strategic directions towards an overall improvement of a school transportation system. It can be a useful decision support tool in the hands of policy and decisionmakers aiming to propose and implement solutions and measures capable of promoting an equitable, safer, and more sustainable school mobility system. Examples of such policies are reflected in measures enhancing public transport access, redesigning and

redeveloping school unit surrounding environments, promoting active transport modes through infrastructure improvements, raising awareness through campaigns and incentives provision, etc. Specifically, the promotion, encouragement, and facilitation, through appropriate interventions, of active (non-motorized) school transport are important for the holistic development of students, fostering physical and mental health, promoting environmental sustainability, and shaping community connections [3,4].

Setting up the appropriate methodology for predicting the transport mode to be used for the completion of everyday school trips demands specific prerequisites to be taken into consideration, including the identification of specific parameters that affect the school travel mode choice. These characteristics encompass individual and socio-demographic features and the specification of attributes relevant to transport modes available, such as travel time, cost, etc.

Taking into consideration the above-mentioned issues, this paper aims to develop a decision support tool for policy making and urban planners, engaging state-of-the-art AI classifiers that assess and quantify the impact of factors affecting the school travel mode choice process. It is structured under four sections. The current introductory section explains the rationale behind the research's general concept and its main objectives and provides a detailed two-fold dimension literature review for uncovering (i) parameters found to affect the school travel mode choice process and (ii) AI classification applications that have demonstrated their ability in predicting attributes associated with human behavioral patterns. Section 2 unfolds the methodology followed. Initially, the data collection and data preparation procedures are described. Following the exploratory and confirmatory factor analysis (EFA and CFA) results, the implementation of the initial ML classification model is presented. Moving on to the deployment and simulation of the ML model, the contribution of each factor on the school travel mode choice is defined and further used in the improvement of a sustainable model choice process. In the next section, three pivotal research questions laying at the heart of the study and already stated clearly in the previous section are investigated and discussed in detail. Lastly, the conclusions section summarizes the outcomes of our work, identifies limitations, deliberately acknowledges challenges, and proposes future work.

*1.2. Research Objectives*

The main objectives of the present paper are as follows:

(a) Development of a solid literature review that: (i) identifies various parameters (observed variables) related to parental behavior characteristics affecting the school travel mode choice process and (ii) presents and describes AI classification applications that have proved their ability in forecasting human behavior characteristics.

(b) Capitalization of research findings from EFA and CFA outcomes from previous research works (by some of the authors of the present paper), with a specific focus on the derived grouped parameters (labeled factors) that exhibiting homogeneous characteristics and interpretive properties in the school travel mode choice process, and aligning these findings with the research scope outlined in (c) to employ ML classification techniques.

(c) Investigation, using ML classification techniques, into the contribution of each labeled factor to the parental school travel mode choice process and identification of factors with the greatest influence on the decision between motorized and non-motorized school travel mode choices.

The current research signifies a groundbreaking and innovative contribution as it marks the inaugural application of ML classification techniques to a questionnaire dataset within the realm of school travel mode choice. To the best of our knowledge, this study stands as the first of its kind to leverage advanced ML methodologies in examining and categorizing qualitative data (parental attitudes, perception, and beliefs) related to the selection of school travel modes. This pioneering approach not only extends the boundaries of current research practices but elicits novel viewpoints for understanding the intricate

dynamics influencing the decision-making process in the context of school transportation choices. The utilization of ML in this domain introduces a novel perspective, shedding light on patterns and insights that were previously unexplored and thereby enriching the discourse on school travel behavior.

*1.3. Literature Review*

The literature review followed a two-fold dimension. At first, it examined and identified the most significant factors that influence the school travel mode choice process. This was fundamental, as most of factors identified were considered for structuring the questionnaire survey used as the main tool for data collection. It also allowed the primary research analysis, where several statistical methods and techniques were adopted leading to the conceptual model and the researchers' hypothesis statement. In parallel, a literature review took place, focusing on AI classification applications that have already proved their ability to forecast human behavior characteristics. Under this scope, the research focused on data collection and the deployment of AI methods, focusing on topics relevant to the transportation and traffic forecasting modeling field. This revealed the limited research on questionnaire-survey-based AI applications in the transportation field generally and in school mobility forecasting more specifically, triggering the authors' interest to implement a state-of-the-art innovative approach for predicting the school travel mode choice.

In relation to the initial segment of the literature review, concise summaries of key findings are delineated below and depicted in Table 1. Notably, the age and gender of students exhibit a direct influence on their choice of school travel mode. Parents of older students state a preference for active transport modes [4–12]. Gender disparities are evident when selecting school travel modes, with girls exhibiting a lower propensity to be engaged in active mobility patterns compared with boys. Pertaining to parental employment, maternal work commitment is associated with an amplifying effect on students being chauffeured to school, while paternal working hours demonstrate no discernible correlation with the decision-making process regarding the school travel mode choice [9,13].

Furthermore, a discernible interconnection is observed between family income and the index of car ownership. Students from affluent households are less inclined to opt for alternative transport modes [14,15], whereas an escalation in the number of vehicles within a household augments the likelihood of students being driven to school [16,17]. The distance between a student's residence and the school unit is identified as a pivotal determinant, whereby an increase in distance diminishes the probability of students utilizing alternative transport modes [14,18]. Concurrently, extended distances are associated with reduced independent school mobility [19–21].

The built environment and parental perceptions of safety within the vicinity of students play pivotal roles in the decision-making process. Factors such as high traffic volumes, elevated speeds, congested road networks with wide axes, and the necessity to navigate overloaded intersections exert a negative influence on parents' decisions to opt for alternative transport modes for their children's school trips [4,7,15]. Interventions such as sidewalk reconstruction and the installation of traffic control devices are identified as facilitators of alternative mobility in areas where such measures are implemented [22]. Similarly, the presence of organized bicycle paths contributes to the preference for bicycles as a mode of school transportation [4].

**Table 1.** Literature findings regarding parents' school travel mode choices.

| Research Work Title | Author(s) and Year | Key Findings |
|---|---|---|
| Active commuting to school: Associations with environment and parental concerns<br>School trips in Germany: Gendered escorting practice<br>A mode choice analysis of School trips in New Jersey<br>Parental attitudes towards children walking and bicycling to school: a multivariate ordered response analysis<br>Integrating parental attitudes in research on children's active school commuting: evidence from a community school travel survey<br>Children's travel: patterns and influences<br>Children's travel behavior a world of difference<br>Children's independent spatial mobility in the urban public realm<br>Statewide prevalence and correlates of walking and bicycling to school | Kerr et al., 2006 [4]<br>Sheiner, 2016 [5]<br>Noland et al., 2010 [6]<br>Seraj et al., 2012, [7]<br><br>Yang and Markowitz, 2012 [8]<br><br>Donald, 2005 [9]<br>Zwerts and Wets, 2006 [10]<br>O' Brien et al., 2000 [11]<br>Evenson et al., 2003 [12] | - School mode choice is directly affected by students' age<br>- Older students are more likely to walk to school than younger students<br>- Older students usually do not travel by car<br>- High school students use the public transport system more often than primary school students<br>- Gender affects parents' decisions in school mobility<br>- Boys are more likely to use alternative transport modes (walking and bicycling)<br>- Girls are more often driven to school than boys |
| Determinants of car travel on daily journeys to school: cross sectional survey of primary school children | DiGuiseppi et al., 1998 [13] | - Parents professional status influences the school mode choice<br>- Maternal employment was found to increase the number of students driven to school<br>- Fathers' working hours are not related to the school mode choice |
| Active transportation mode choice behavior across genders in school trips<br>Intra-household travel interactions, the built environment and school travel mode choice: an exploration using spatial models<br>An analysis of the determinants of children's weekend physical activity participation<br>Letting children be free to walk | Ermagun and Samimi, 2012 [14]<br>Mitra and Buliung, 2012 [15]<br><br>Copperman and Bhat, 2007 [16]<br>Mackett, 2011 [17] | - Students from families with high incomes are less likely to use alternative transport modes<br>- An increase in the number of vehicles available in a household increases the number of students driven to school |
| Children's travel behavior and its health implications<br>Children's independent mobility to school, friends and leisure activities<br>Parental chauffeurs: what drives their transport choice<br>Built environment and children's travel to school<br>Evaluation of the California safe routes to school legislation: urban form change and children's active transportation to school | Mackett, 2013 [18]<br>Fyhri and Hjorthol, 2009 [19]<br>Carver et al., 2013 [20]<br>Curtis et al., 2015 [21]<br>Boarnet et al., 2005 [22] | - Long distances between students' residences and school units reduce the possibility for students to walk to school<br>- Long distances are associated with less independent mobility<br>- Built environment plays a critical role in school mode choice<br>- High traffic loads, high speeds, congested road networks, and overloaded intersections negatively affect parents in choosing alternative transport modes<br>- Reconstruction of sidewalks and installation of traffic control devices enhance alternative mobility |

**Table 1.** *Cont.*

| Research Work Title | Author(s) and Year | Key Findings |
|---|---|---|
| Barriers to children walking to or from school<br>Why parents drive children to school: implications for safe routes to school programs | Martin and Carlson, 2005 [23]<br>McDonald and Aalborg, 2009 [24] | - Criminality greatly influences parents school mode choice<br>- Bullying, gang activity, and the chance of abduction are parents' main fears and concerns for not letting their children walk or bike to school<br>- Girls are less likely to walk to school in a neighborhood where the levels of safety are low |

Safety emerges as a paramount factor in the decision-making process for school travel modes. Concerns and anxieties regarding the perceived safety levels in school transportation are recurrent themes in numerous studies [23,24]. However, it is noteworthy that fatal pedestrian and cyclist accidents during school hours exhibited a marked decline from 1987 to 2009 [25]. Conversely, parental apprehensions about criminality are grounded more in societal norms than actual risks. Kidnapping incidents account for only 2% of violent crimes involving young people and 4% of all kidnappings in school districts [26].

Regarding the use of AI approaches, a remarkable interest has been noticed within recent years in a number of different research fields; medicine, healthcare, phycology, transportation, etc., are strong examples of the fields where these approaches mostly apply [27]. Within the domain of transportation, the utilization of artificial intelligence (AI) applications for traffic forecasting models represents a cutting-edge and technologically advanced approach [28–35]. This innovative approach has the potential to alleviate uncertainties during the planning and design phases of future transportation infrastructure and investments [36,37]. By doing so, it addresses challenges associated with traffic congestion and issues within transportation networks [38,39]. Furthermore, it contributes to mitigating the environmental footprint of transportation [40].

The methods employed for data collection and research analysis needs vary, encompassing popular approaches such as crowdsourcing platforms, private collection by AI developers, pre-cleaned and pre-packaged data, automated data collection, and generative AI [41–43].

The contribution of AI approaches has also become particularly significant in the processing of data originating from survey research. In these surveys, the recorded opinions of users regarding transportation infrastructures or services constitute the input data for AI approaches, while preferences towards or satisfaction with the transportation infrastructure or service represent the output data.

In a more detailed examination, an exploration into the feasibility of predicting the perceived quality of public transport services, as perceived by users, has been undertaken. This investigation relies on artificial intelligence (AI) models trained with data derived from a questionnaire survey gauging 655 users' perceptions of urban bus services in Dhaka, the capital and largest city of Bangladesh [44]. Out of twenty-two selected service quality features, the most pivotal characteristics were systematically ranked based on their influence on user decision-making procedures regarding public transport utilization. The AI models, subsequent to training, forecasted that punctuality, reliability, service frequency, seat availability, and travel experience were the most critical determinants. Additionally, an AI model specifically designed for forecasting the quality of public transport services in rural areas was developed, leveraging 401 users' perceptions obtained through a questionnaire survey. Thirteen indicators were considered, yielding satisfactory predictive capabilities for user dissatisfaction levels related to the reliability of the public transport system and the availability of seats within the buses [45].

Deb et al. [46] validated artificial neural networks (ANN) for modeling public transportation service quality. Focused on Granada's metropolitan bus service, the study analyzed data from a survey conducted with 858 passengers. The results highlighted significant differences in attribute categories, showing frequency as the most influential factor in service quality determination. Other attributes, including speed, information, and proximity, were also identified as impactful contributors. The findings contribute valuable insights into enhancing public transportation planning and service design for improved passenger satisfaction.

Further application of AI was observed in the investigation of modal shifts from private vehicles to public transport following the introduction of smart transport services for commuters in Mersin city, located in southern Turkey [47]. Utilizing data obtained from two questionnaire surveys (comprising a total sample of 606 participants) and employing algorithmic procedures to calculate the shortest route, the researchers identified trip characteristics that facilitated the forecasting of the percentage of transport mode shifts after

the implementation of smart services. The study concluded that artificial neural networks (ANN) were apt for modeling dynamic transport systems and forecasting modal shifts.

The determination of factors influencing school trip mode choice was the focus of a study in Kandy, Sri Lanka [48]. Findings from a questionnaire survey to 1983 students (aged 6–13 years) revealed that gender, age, household income, school type, and distance played significant roles in shaping school mobility patterns. However, the study acknowledged limitations in generalizability to other case studies due to varying socio-economic and weather conditions.

Trying to explain and predict the school-goers mode choice behavior, the data collected through a questionnaire survey (sample of 2747 students) was analyzed (1484 valid responses) under the employment of three different machine learning tools (MLT). The data concerned public schools in the Al-Khobar and Dhahran cities in the Kingdom of Saudi Arabia. This research work concluded that travel time, family income, and parental education level were the prime variables to dictate the travel mode choice behavior [49].

Table 2 summarizes research works undertaken regarding application of AI in the transportation field. The literature review highlighted a scarcity of research on school travel mode choice models, particularly in the realm of AI applications. Consequently, the primary research objectives were formulated to establish a logical framework for a more comprehensive understanding of human behavior pattern recognition in school travel mode choices, incorporating AI applications. This framework emphasized the initiation of a scientific tool to design and implement sustainable urban policies. Within this context, the prediction of school travel mode choices, based on both quantitative and qualitative data obtained through a dedicated questionnaire survey targeting students' parents, serves as a pathway to explore new computational intelligence capabilities and chart new directions for the overall optimization of the future school transportation system.

**Table 2.** Literature findings regarding the deployment of AI applications in the transport sector.

| Research Work Title | Author/Year | Research Topic | Method(s) Used |
|---|---|---|---|
| Traffic flow analysis based on the real data using neural networks | Pamula, 2012 [28] | Traffic flow forecast | ANN models |
| Forecasting demand for low-cost carriers in Australia using an artificial neural network approach | Srisaeng and Baxter, 2015 [29] | Air transport demand forecast | Econometric and ANN models |
| Comparative analysis for traffic flow forecasting models with real-life data in Beijing | Rong et al., 2015 [30] | Traffic flow forecast | Autoregressive integrated moving average (ARIMA) and ANN models |
| Deep trend: A deep hierarchical neural network for traffic flow prediction | Dai et al., 2017 [31] | Traffic flow time series | DeepTrend ANN models |
| Predicting the daily traffic volume from hourly traffic data using artificial neural network | Siddiquee and Hoque, 2017 [32] | Traffic flow forecast | ANN models |
| A deep learning approach for short-term airport traffic flow prediction | Yan et al. [33] | Air transport demand forecast | Machine learning techniques (gated recurrent units, graph convolutional networks, ANN) |
| Real–time intraday traffic volume forecasting—A hybrid application using singular spectrum analysis and artificial neural networks | Kolidakis et al., 2019 [35] | Traffic flow forecast | Hybrid singular spectrum analysis (SSA)—ANN models |
| Short-term traffic forecasting: Where we are and where we are going | Vlahogianni et al., 2014 [36] | Traffic flow forecast | Review on short-term traffic forecasting models |
| Evaluating toll revenue uncertainty using neural network models | Zhao and Zhao, 2017 [37] | Traffic demand forecast | Big data-based models |
| A prediction model based on time series data in Intelligent Transportation System | Wu et al., 2013 [38] | Massive traffic transportation network problems | ARIMA and generalized regression neural network (GRNN) models |
| Comparative Traffic Flow Prediction of a Heuristic ANN Model and a Hybrid ANN-PSO Model in the Traffic Flow Modelling of Vehicles at a Four-Way Signalized Road Intersection | Olayode et al., 2021 [39] | Short-term urban traffic flow forecast | Heuristic ANN and hybrid ANN–particle swarm optimization models |
| Modeling of CO emissions from traffic vehicles using artificial neural networks | Azeez et al., 2019 [40] | Vehicular carbon monoxide emissions | Hybrid geographic information system and ANN model |
| Identifying and describing streets of interest | Skoutas et al., 2016 [42] | Streets of interest identification | Multiclass support vector machines (SVMs) classification |
| Bus service quality prediction and attribute ranking: a neural network approach | Islam et al., 2016 [44] | Evaluation of public transport service quality | GRNN, probabilistic neural network, pattern recognition neural network models |
| Neural networks approach for evaluating quality of service in public transportation in rural areas | Wagale et al., 2014 [45] | Evaluation of public transport service quality | ANN models |
| Service quality estimation and improvement plan of bus Service: A perception and expectation based analysis | Deb et al. (2022) [46] | Evaluation of public transport service quality | ANN models |
| Prediction of modal shift using Artificial Neural Networks | Akgol et al. (2014) [47] | Shift from private vehicles to public transport | ANN models |
| Exploring home-to-school trip mode choices in Kandy, Sri Lanka | Dias et al., 2022 [48] | School travel mode prediction | ANN models |
| Travel-to-school mode choice modelling employing Artificial Intelligence techniques: a comparative study | Assi et al., 2019 [49] | School travel mode prediction | Machine learning tools (extreme learning machine, SVM, ANN) |

## 2. Methodology and Experimentation

The methodology of the proposed paper includes 6 steps, which are described in the following sections (Figure 1).

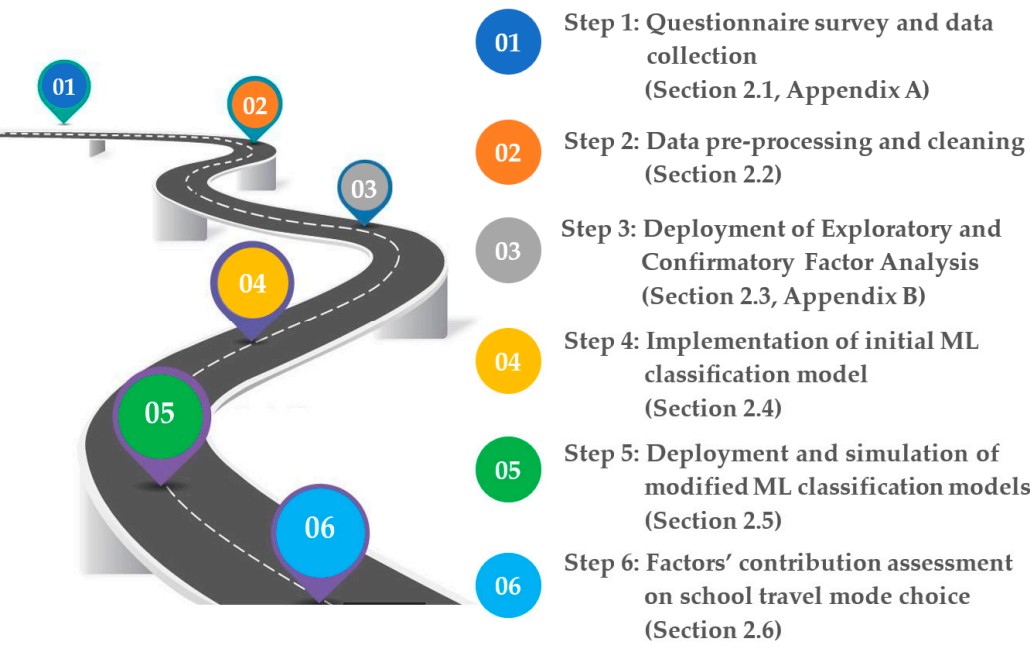

**Step 1: Questionnaire survey and data collection (Section 2.1, Appendix A)**

**Step 2: Data pre-processing and cleaning (Section 2.2)**

**Step 3: Deployment of Exploratory and Confirmatory Factor Analysis (Section 2.3, Appendix B)**

**Step 4: Implementation of initial ML classification model (Section 2.4)**

**Step 5: Deployment and simulation of modified ML classification models (Section 2.5)**

**Step 6: Factors' contribution assessment on school travel mode choice (Section 2.6)**

**Figure 1.** Methodology roadmap.

### 2.1. Questionnaire Survey and Data Collection

For the purpose of primary research and data acquisition, a questionnaire was meticulously formulated following a thorough analysis presented in the preceding literature review. The questionnaire comprised three distinct sections (Appendix A): the initial section encompassed inquiries pertaining to respondents' socio-economic and demographic attributes (Q1 to Q9), whereas the second section delved into aspects related to the completion of school trips (Q10 to Q16). The third section consisted of three parts: the first part presented eighteen pivotal features influencing parents in the travel mode choice decision-making process to ascertain their significance level (Q17). Subsequently, the second part scrutinized the role of the environmental structure within which students undertook their school travels (Q18). In the specific part, parents were required to express their level of agreement or disagreement concerning thirteen statements delineating the environment, including the route from their residence to the school unit. The third part of the questionnaire (Q19) focused on exploring fifteen statements pertaining to parents' travel patterns, beliefs, and attitudes to discern the impact of their perceptions regarding various transport modes on the school travel mode choice process.

The questionnaire survey was conducted in the city of Thessaloniki, located in northern Greece, the second-largest city in the country with an approximate population of one million residents and 100,000 primary and secondary school students. The data collection spanned from May to June and from September to November 2019. The determination of the minimum sample size adhered to the following equation [1]:

$$n \geq N \cdot \left[ 1 + \frac{N-1}{p \cdot (1-p)} \cdot \left( \frac{d}{z_{a/2}} \right)^2 \right]^{-1} \tag{1}$$

for N = 100,000 school students, $p$ = 50%, d = ±5%, and $z_{\alpha/2}$ = 1.96 for a confidence level of 95%. Based on Equation (1) and for the case examined, at least 383 questionnaires were required to be completed. However, in total 512 questionnaires were collected, with 496 of them deemed suitable for machine learning (ML) classification training and modeling due

to their completeness and correctness. The questionnaire completion process involved a dual approach: in-person interviews were conducted and parents were also encouraged to complete the questionnaire online through a Google Docs format file distributed to them via e-mail.

In terms of the respondents' profile, the analysis of the first part of the questionnaire revealed the following results [1]: 33.5% were men and 66.5% were women. The prevalence of women over men in the sample aligns with the relevant literature, as primarily women (up to 79%) actively participated in organizing their children's school commute [50]. Nearly 70% of the respondents fell within the age range of 40–49 years old, and 9 out of 10 were married (91.0% in Greece in 2019). Concerning educational level, 35.2% held a bachelor's degree. In terms of employment status, 85.4% were full-time employees (82.7% in Greece in 2019). Eight out of ten parents possessed a driving license, and the mean value for the car ownership index was estimated at 1.55 vehicles per household (1.50 in Greece in 2019). Regarding the students' gender, 54.1% were girls and 45.9% were boys.

### 2.2. Data Pre-Process and Cleaning

Data pre-processing refers to the initial stage of data preparation, which involves transforming and structuring the data to make them suitable for analysis. This can include tasks such as data integration, data normalization, and data transformation. Data cleaning involves identifying and correcting errors, inconsistencies, and inaccuracies in the data. This can include tasks such as removing duplicate records, filling in missing values, and correcting spelling errors or typos. Overall, data pre-processing and data cleaning are important steps in preparing data for analysis, as they ensure that the data is accurate, consistent, and structured in a way that can be easily analyzed [51].

In our specific context, the meticulous process of data pre-processing and cleaning has resulted in a significant refinement of our dataset. From the initial pool of 512 questionnaires, our efforts have led to the retention of 496 questionnaires that are now not only complete but also free from any data inconsistencies or errors. These high-quality, thoroughly curated questionnaires serve as the foundation for our ML classification training and modeling endeavors. This precise data preparation ensures that the insights and patterns extracted from our dataset are accurate and reliable, paving the way for robust and effective ML models. Our commitment to data quality and integrity is a fundamental step in the journey of harnessing the power of data science to drive meaningful and precise outcomes in the specific domain.

### 2.3. Results of Exploratory and Confirmatory Factor Analysis

The analytical description and evaluation of EFA and CFA processes, along with the calculation of appropriate metrics, indices, and matrices, were conducted and published by some of the authors in prior research [1,52,53]. Therefore, these processes are not reiterated in the paper to avoid gratuitous redundancies. Based on these findings, nine factors were extracted through the EFA (Appendix B), accounting for 61% of the total variance in questionnaire parameters (observed variables), as described below:

- Factor MOTMODE (mean value in 5-point Likert scale: 3.84, the qualitative interpretation of factors' mean values refers to the last column of the Table A2, Appendix B) covers some objective parameters of parental motivation to choose a transport mode (student safety and convenience, travel time and distance between residence and school unit, student age, working hours of parents, and the possibility for someone else, other than the respondent parent, to assist with school transportation).
- Factor MOTHEALTH (mean value: 3.39) includes the parameters of parental motivation to choose a transport mode related to the physical and mental health of a student. A possible increase in the significance that parents attach to the specific parameters is expected to further increase the use of alternative transport modes such as bicycle and walking.

- Factor ATTBUS (mean value: 2.26) involves the parameters related to the parents' perception regarding the use of public buses. In case of increasing the level of agreement parents declare in the statements (parameters composing this factor), a reduction in the use of walking and bicycling in favor of a public bus is expected to be noticed.
- Factor ATTCAR (mean value: 3.06) contains the parameters related to the parents' perception regarding the use of a private vehicle. The increase in the level of agreement parents declare for these statements is estimated to reduce the use of walking and bicycling in favor of motorized transport modes.
- Factor ATTWALKBIKE (mean value: 4.14) includes the parameters related to the parents' perception regarding the use of non-motorized transport modes such as walking and bicycle. In case of increasing the level of agreement in the specific parameters that compose this factor, it is estimated that parents will prefer their children to walk or bike to school, in a larger degree, since they have already shaped a positive attitude towards the use of non-motorized transport modes.
- Factor MOTCAR (mean value: 3.07) consists of parameters related to the possibility of using a private vehicle. An increase in the level of significance that parents attribute to the specific parameters is expected to cause a negative attitude towards the use of non-motorized transport modes.
- Factor NEIGBSAF (mean value: 3.29) encompasses parameters associated with the perceived level of security within the neighborhoods along the route connecting students' residences and the school units. Increasing the level of security sense parents perceive regarding the neighborhood students cross on their trip from the school unit to their residence and vice versa is expected to act in favor of walking and bicycling in their final mode choice.
- Factor ROUTESAF (mean value: 2.30) contains the parameters related to safety perception facilitated by the sidewalks and the whole path the student follows. An increase in the level of agreement parents declare for these statements representing the specific parameters is expected to also increase the use of non-motorized transport modes, as these parameters characterize an integrated and well-organized (in terms of infrastructure) built environment shaped to act in favor of walking and bicycling.
- Finally, factor MODE comprises the actual and preferable parental mode choice regarding the school transportation.

We notice that from factor analysis, one factor (MOTMODE) emerged with rather heterogeneous questionnaire parameters (observed variables). Two factors (NEIGBSAF and ROUTESAF) were also identified, encompassing observed variables related to the evaluation of parents regarding the safety and security sense provided by the surrounding environment of neighborhoods, sidewalks, and the entire path between the residence and the school unit. Finally, five factors (MOTHEALTH, ATTBUS, ATTCAR, ATTWALKBIKE, and MOTCAR) are primarily associated with parental perceptions and beliefs regarding the use and usability of various transport modes (including private vehicles, public buses, walking, and bicycling).

As stated in the description of each factor, changes in parents' perceptions or beliefs regarding the level of significance or the level of agreement they attribute to the various parameters (that constitute each factor) are expected to affect parental attitudes positively or negatively towards the use (or non-use) of non-motorized transport modes. We emphasize and focus on the distinction between motorized and non-motorized school commuting, considering that, as highlighted in the introductory part of the study (Section 1.1), the choice of non-motorized transport modes incorporates individual, environmental, and social benefits [3,4].

In Figure 2, the percentages are provided as derived from the questionnaire survey on the use of each transport mode during school commuting. It is observed that non-motorized transportation is preferred by 66.1% of the surveyed parents, compared with a percentage of 33.9% who preferred a motorized mode for their child's daily school commuting.

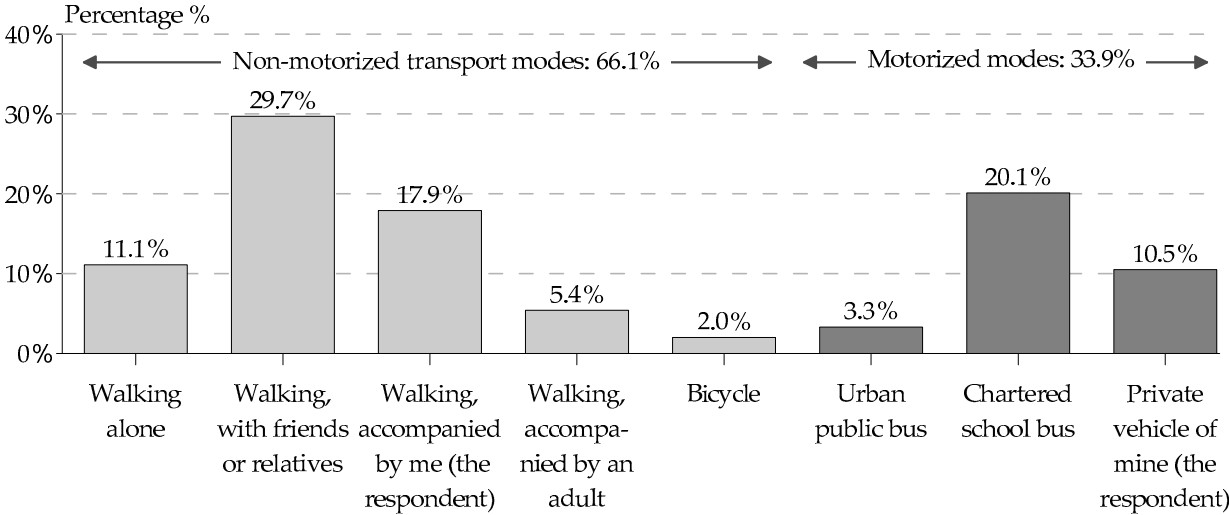

**Figure 2.** School transport mode preferred by parents for their children.

Therefore, the following research questions arise:

1. Can AI classification techniques, by using parents' responses to various questions (composing the observed variables) in the survey, predict the parental choices regarding motorized and non-motorized school transportation and to what level of accuracy?

2. Which of the two factors (NEIGBSAF and ROUTESAF), which can directly (short and medium term) be influenced by state or/and municipal authorities through improvement and rehabilitation projects, has the greatest influence on encouraging non-motorized school transportation?

3. Which of the five factors (MOTHEALTH, ATTBUS, ATTCAR, ATTWALKBIKE, and MOTCAR) related to parental perceptions and beliefs, which can only indirectly (along term) be influenced by the state (through information and awareness-raising campaigns, incentives provision, etc.), has the greatest impact on encouraging non-motorized school travel?

Answers to the above three questions will be attempted in Section 3 of the paper.

*2.4. Implement of the ML Classification Model*

The ML part of this work encompasses the development of a ML classification that, upon training on the questionnaire survey and parents' responses to various questions (composing the observed variables) in the survey, can be utilized to make predictions regarding the parental decision between motorized and non-motorized transport modes for their child's daily school commuting (Figure 3).

The Pearson correlation coefficient (PCC) matrix is expressed as a heatmap (Figure 4). The PCC can describe the linear correlation between two features [54]. We find that some features have a stronger linear correlation with other features, which empowers the correlations' analysis through the application of EFA and CFA that took place, leading to factor clustering and labeling (Appendix B).

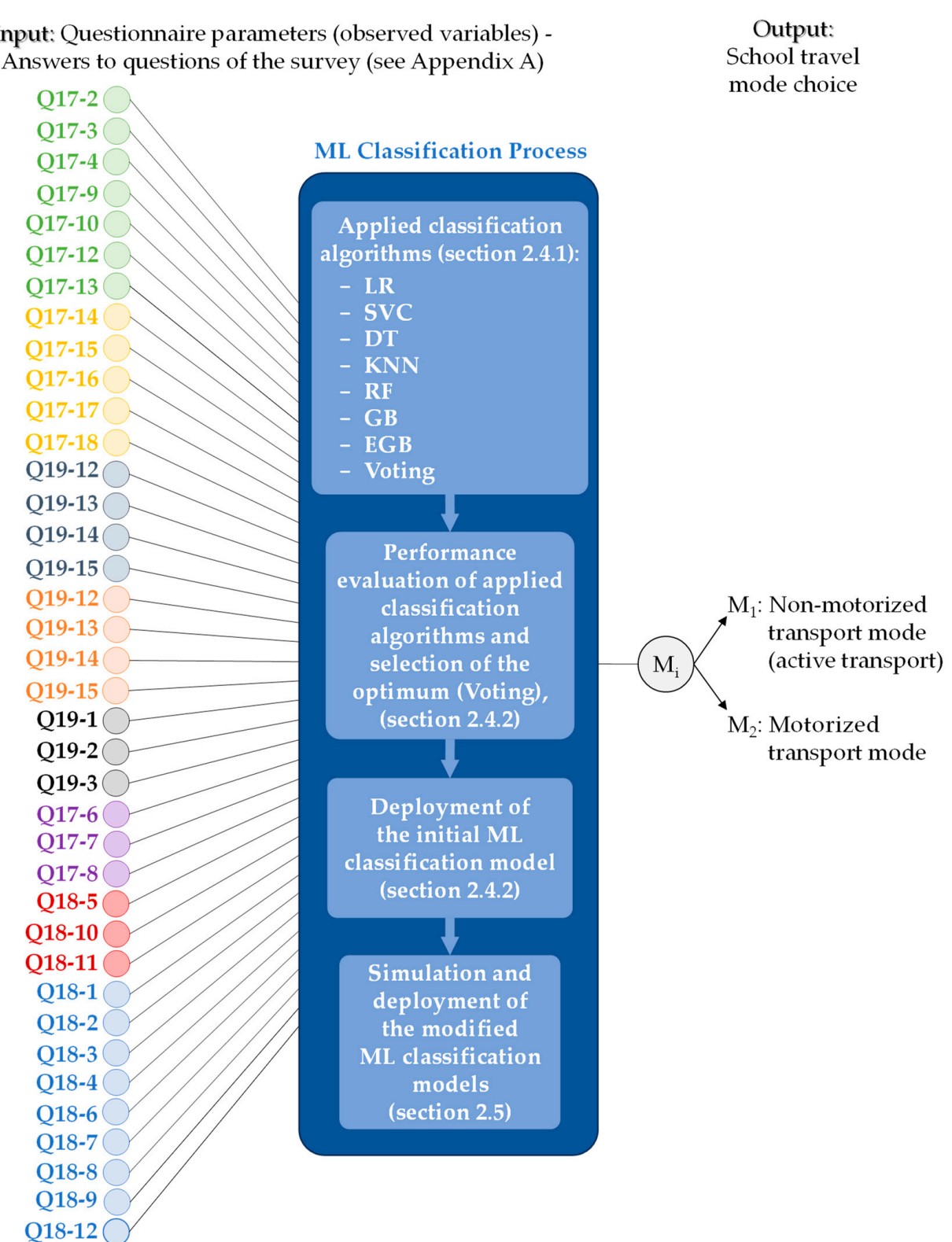

**Figure 3.** The schematic presentation of the ML classifier. Upon training on the parents' responses to questionnaire survey predicts the related to the school travel mode choice decision.

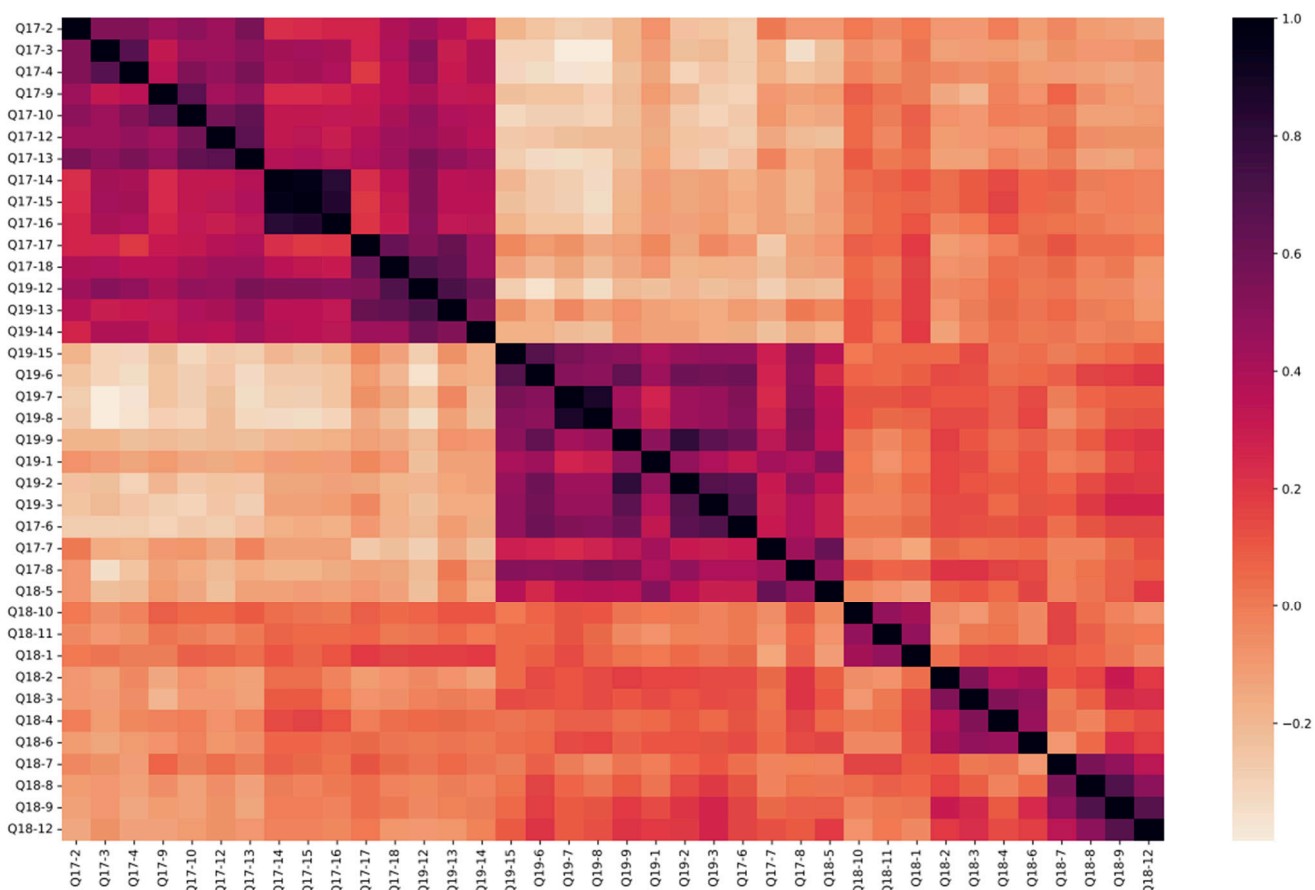

**Figure 4.** Heatmap of Pearson correlation coefficient matrix.

2.4.1. Selection of Classification Algorithm

To this end, a series of seven classifiers employing various methodologies were tested (including logistic regression, K-nearest neighbor, decision tree, support vector machines, random forest, gradient boosting, and extreme gradient boosting) and trained with default parameters (i.e., without fine-tuning). More particularly:

- Logistic Regression (LR) [55]: a classification algorithm that employs a linear function that is formulated by aggregating weighted input data features. The main objective is to optimize these feature weights, so that a predefined cost function (such as mean squared error) is minimized.
- Support Vector Classifier (SVC) [56]: a model whose primary goal is to find the optimal separating hyperplane to distinguish among different classes, specifically the one that maximizes the margin or distance from the hyperplane to the nearest data points of each class.
- Decision Trees (DT) [57]: a tree-based classification method. The DT construction involves two main processes: tree building, which progressively splits records based on certain criteria, and tree pruning to control tree complexity by reducing the number of leaf nodes.
- K-Nearest Neighbor (KNN) [58]: an approach according to which test instances are classified by measuring their similarity/distance to training instances, typically using similarity or distance metrics such as Euclidean distance and cosine similarity. The KNN is then determined based on their proximity to the test instance and the class of the test instance is assigned based on majority voting among these K-neighbors.
- Random Forest (RF) [59]: an ensemble learning method that fits multiple deep decision trees to different subsamples of a given dataset. The predictions of these decision trees are then combined to improve the overall performance and avoid overfitting.

- Gradient Boosting (GB) [60]: a stagewise additive model where an ensemble of prediction models is built by incrementally adding weak learners. The goal is to minimize the loss of the ensemble by employing a gradient-descent-like optimization procedure that adjusts the predictions of each weak learner in the ensemble to reduce loss.
- Extreme Gradient Boosting (XGB) [61]: a classification method that constructs an ensemble of decision trees, training them sequentially. It calculates the gradient of the loss function to determine how the model's parameters should be adjusted to reduce errors. As a result, each new tree corrects the errors of the previous ones.

### 2.4.2. ML Classification Model Training and Testing

Initially these classifiers were trained with default parameters (i.e., without fine-tuning). This involves dividing the dataset into a training set and a validation set and then using the training set to teach the model how to classify the data. This training was performed using 10-fold cross-validation and then the performance was assessed to select the top three performing classifiers based on their performance metrics (Table 3).

**Table 3.** Classifiers performance.

| Classification Model | Accuracy | F1–Score | Precision | TPR (Recall) | AUC–ROC Score | FPR | MCC | CCI | ICI |
|---|---|---|---|---|---|---|---|---|---|
| LR | 0.71 | 0.72 | 0.68 | 0.76 | 0.71 | 0.33 | 0.43 | 106 | 43 |
| SVC | 0.89 | 0.89 | 0.89 | 0.87 | 0.89 | 0.10 | 0.77 | 132 | 17 |
| DT | 0.86 | 0.86 | 0.82 | 0.90 | 0.86 | 0.18 | 0.72 | 128 | 21 |
| KNN | 0.79 | 0.77 | 0.80 | 0.75 | 0.79 | 0.17 | 0.58 | 118 | 31 |
| RF | 0.89 | 0.89 | 0.89 | 0.89 | 0.89 | 0.10 | 0.78 | 133 | 16 |
| GB | 0.90 | 0.90 | 0.91 | 0.87 | 0.89 | 0.08 | 0.79 | 134 | 15 |
| XGB | 0.87 | 0.87 | 0.88 | 0.83 | 0.86 | 0.10 | 0.73 | 129 | 20 |
| Voting | 0.90 | 0.90 | 0.88 | 0.92 | 0.90 | 0.11 | 0.80 | 134 | 15 |

Next, the top-performing classifiers underwent hyperparameter fine-tuning to optimize their predictive performance. This fine-tuning process employed a random 30% of the data for testing and the remaining 70% for training. In addition to the best-performing fine-tuned classifier, a soft voting ensemble classifier (hereafter voting classifier) was constructed by averaging the prediction of the three individual fine-tuned classifiers. The aim was to leverage their combined predictive capabilities. Subsequently, this ensemble classifier was tested and achieved competitive results, although the three individual fine-tuned classifiers were slightly inferior to the top-performing classifier (Table 3). It is important to highlight that, except for the KNN and LR classifiers, all other classifier models demonstrated consistent and robust performance, surpassing an 80% accuracy threshold (Table 3). This remarkable level of performance renders them exceptionally well suited for handling various classification tasks within the scope of our research.

Based on the models' assessments using a variety of diverse evaluation metrics, it is evident that the GB and the voting classifier exhibit the best performances among the models tested. Although these performances are competitive, we end up selecting the voting classifier as the best model for our application due to the higher scores regarding TPR (true positive rate, also known as recall), AUC–ROC score (area under the curve of the receiver operating characteristic), and MCC (Matthew's correlation coefficient) that the latter demonstrates.

In particular, MCC is a more reliable statistical metric that produces a high score only if the prediction obtained good results in all of the four confusion matrix categories, namely true positives, false negatives, true negatives, and false positives, proportionally both to the size of positive elements and the size of negative elements in the dataset. Moreover, MCC remains invariant to the selection of the positive class, thereby ensuring the robustness of the metric in the event of the class imbalance that we encounter in the data used in this work [62].

As for the AUC–ROC score, elevated scores indicate higher values for both TPR (recall) and FPR (false positive rate), resulting in this way a greater degree of balance between them, which is suitable for quantifying the effectiveness of the selected model in addressing the issues arising from class imbalance.

Finally, the higher recall scores, which can be considered as the accuracy on the positive class (i.e., use of a non-motorized transport modes), highlight the efficiency of the voting classifier in correctly classifying the positive class instances. This is crucial, given that the cost of false negatives outweighs the cost of false positives, since the ultimate goal is to capture the majority, if not all, of the instances opting for the use of non-motorized transportation modes for the school trip instead of motorized ones so that appropriate policies can be applied by the local authorities.

In terms of the scores of the CCI (correctly classified instances) and ICI (incorrectly classified instances), both GB and voting classifier present similar performances.

Following the successful completion of the optimization process, we identified the voting classifier (the hybrid classifier that combined the SVC, RT, and XGB classifiers) as the top-performing classifier. This classifier, coupled with its finely tuned hyperparameter configuration, emerged as the standout choice. A confusion matrix of the chosen model summarizes the classifier performance (Table 4), with insights into its error types.

**Table 4.** Confusion matrix for the selected classification model.

|  |  | Mode for school commuting predicted by the classifier | |
| --- | --- | --- | --- |
|  |  | Motorized transport mode | Non-motorized transport mode |
| Mode for school commuting preferred by the parents | Motorized transport mode | 47 75.8% | 15 24.2% |
|  | Non-motorized transport mode | 6 6.9% | 81 93.1% |

*2.5. Deployment and Simulation of Modified ML Classification Models*

As mentioned in Section 2.4.2, the selected voting classifier was successfully utilized upon training on the questionnaire survey and parents' responses to various questions to make predictions on parental school travel mode choice and especially their decision between motorized and non-motorized transport modes for their child's daily school commuting.

However, as mentioned in the research questions at the end of Section 2.3, it would be particularly interesting and useful to evaluate the impact of changes in each of the seven labeled factors (MOTHEALTH, ATTBUS, ATTCAR, ATTWALKBIKE, MOTCAR, NEIGBSAF, and ROUTESAF) related to parental perceptions and beliefs (as recorded through the relevant questions in Appendix A) on shaping the parental preference ratio between non-motorized (66.1%) and motorized (33.9%) school transport modes. The process followed is described below:

i.    The factor under assessment was selected, e.g., factor ROUTESAF.

ii.    The questionnaire parameters (observed variables) that compose the ROUTESAF factor were isolated. In this case, questions Q18-1, Q18-2, Q18-3, Q18-4, Q18-6, Q18-7, Q18-8, Q18-9, and Q18-12 (see Appendix A).

iii.    A process of increasing (and also decreasing on a next step) the initial degrees of agreement/disagreement of the participants in the questionnaire survey regarding statements Q18-1, Q18-2, Q18-3, Q18-4, Q18-6, Q18-7, Q18-8, Q18-9, and Q18-12 was initiated. During the increase in the degree of agreement, respondents with the lowest degree of agreement in each statement were identified and the increase process started from there. Thus, responses such as "Strongly disagree" were initially transformed into "Disagree", and when these were completed, "Disagree" was then transformed into "Undecided" and so on. Similarly, during the decrease in the degree of agreement, respondents with the highest degrees of agreement ("Strongly agree" and "Agree") in

each statement were identified and the process of decreasing degrees of agreement started also from there.

iv. For the new mean value corresponding to the factor ROUTESAF, after the modification of the previous step, the top-performing classification model was applied to predict the new percentage of non-motorized school transport modes based on the modified mean value of the factor.

v. This process identified several pairs of values (mean value of the ROUTESAF factor, percentage of non-motorized school transport modes). Figure 5 illustrates the change in the percentage of non-motorized school transport modes based on the change in the initial degrees of agreement/disagreement of the parents participating in the questionnaire survey regarding statements of this specific factor.

vi. All responses were then reset to the initial values, as recorded in the questionnaire survey, another factor was selected, and the process repeated from the beginning.

Figure 6 illustrates the impact of changes in mean values of each one of the other six other factors (MOTHEALTH, ATTBUS, ATTCAR, ATTWALKBIKE, MOTCAR, and NEIGBSAF) on the parental preference ratio between non-motorized and motorized school transport modes. Non-linearities between the percentage of choice for non-motorized school commuting are evident from Figures 5 and 6, while the changes in the quality of pedestrian and cyclist infrastructure as well as the changes in parental perceptions and beliefs (regarding environmental and economic benefits, children's health and socialization, urban environmental sustainability, etc., resulting from non-motorized commuting) are noticeable. However, these non-linearities enhance the advantage of using ML classification models that are not only suitable but also suggested for the analysis and interpretation of non-linear phenomena [45].

### 2.6. Factor Contribution Assessment for School Travel Mode Choice

Table 5 quantifies what was graphically depicted in Figures 5 and 6, that is the impact of changes in mean values of each factor on the parental preference ratio between non-motorized and motorized school transport modes.

As already noted, non-linear correlations are observed between the percentage of non-motorized school commuting and the changes in the values of the factors influencing it. However, an approach can be attempted regarding the number of parents opting for non-motorized commuting (or the opposite) depending on the changes in the mean values of the factors.

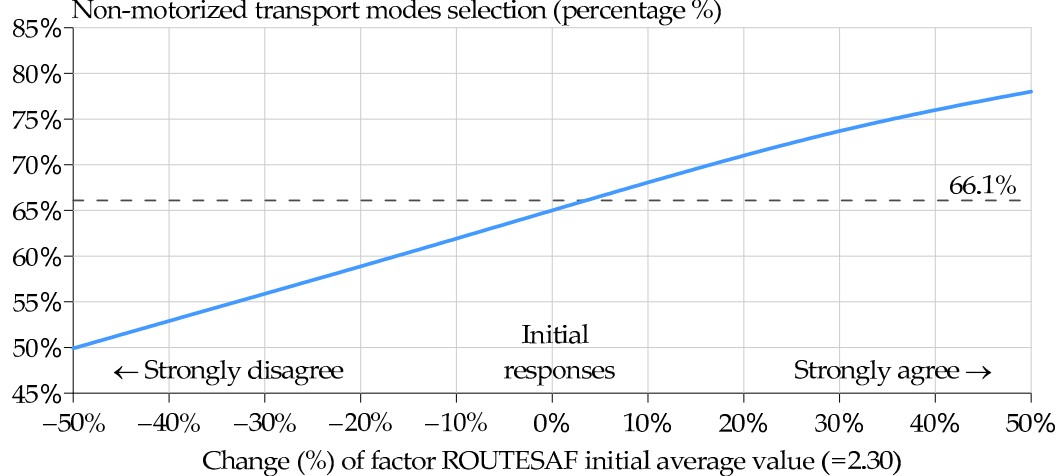

**Figure 5.** Change (%) in the percentage of non-motorized school transport modes based on the change (%) in the initial degree of agreement/disagreement of the parents in the questionnaire survey regarding statements of the factor ROUTESAF.

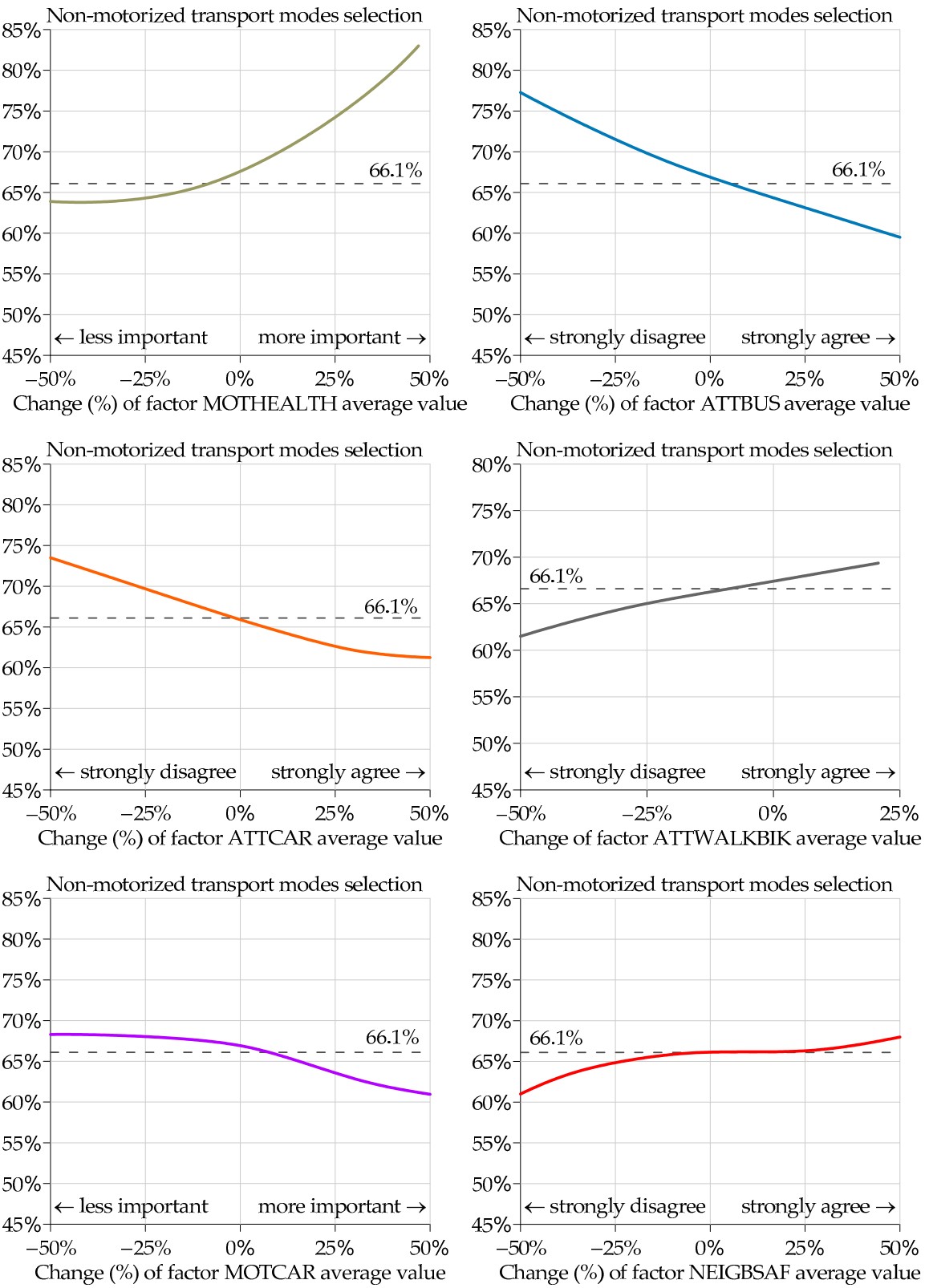

**Figure 6.** Change (%) in the percentage of non-motorized school transport modes based on the change (%) in the initial degree of agreement/disagreement of the parents in the questionnaire survey regarding statements of the factors MOTHEALTH, ATTBUS, ATTCAR, ATTWALKBIKE, MOTCAR, and NEIGBSAF.

**Table 5.** Impact of changes in mean values of each factor on the parental preference for non-motorized school transport modes.

| Labeled Factors (Latent Variables) | Change (%) and Modified Values of Factors' Means in the 5-Point Likert Scale | | | | | | Percentage (%) of Non-Motorized School Transport Modes for Specific Changes of Factors' Mean Values | | | | | |
|---|---|---|---|---|---|---|---|---|---|---|---|---|
| | −40% | −20% | 0% | +20% | +40% | +50% | −40% | −20% | 0% | +20% | +40% | +50% |
| MOTHEALTH: Parameters of parental motivation to choose a transport mode related to the physical and mental health of a student | 2.04 [1] | 2.72 | 3.39 | 3.90 | 4.41 | >5.00 | 63.8% | 64.6% | 66.1% | 72.7% | 79.7% | n.a. |
| ATTBUS: Parents perception regarding the use of public buses | 1.36 [2] | 1.81 | 2.26 | 2.60 | 2.94 | 3.39 | 74.9% | 70.6% | 66.1% | 64.0% | 61.0% | 59.6% |
| ATTCAR: Parents perception regarding the use of private vehicles | 1.84 [2] | 2.45 | 3.06 | 3.52 | 3.98 | 4.59 | 71.9% | 69.0% | 66.1% | 63.3% | 61.5% | 61.3% |
| ATTWALKBIK: Parents perception regarding the use of non-motorized transport modes (walking, bicycle) | 2.48 [2] | 3.31 | 4.14 | 4.76 | >5.00 | >5.00 | 63.2% | 65.7% | 66.1% | 69.4% | n.a. | n.a. |
| MOTCAR: Parameters related to the usability of private vehicles | 1.84 [1] | 2.46 | 3.07 | 3.53 | 3.99 | 4.61 | 68.3% | 67.9% | 66.1% | 64.4% | 61.7% | 60.9% |
| NEIGBSAF: Parameters related to the sense of security provided by the neighborhood | 1.97 [2] | 2.63 | 3.29 | 3.78 | 4.28 | 4.94 | 62.9% | 65.1% | 66.1% | 66.2% | 67.0% | 67.8% |
| ROUTESAF: Parameters related to the sense of safety provided by the sidewalks and the whole path the student follows | 1.38 [2] | 1.84 | 2.30 | 2.65 | 2.99 | 3.45 | 52.8% | 58.8% | 66.1% | 71.1% | 76.0% | 77.9% |

[1] 1: Not important at all, 5: Absolutely important. [2] 1: Strongly disagree, 5: Strongly agree. n.a.: not applicable.

In an attempt to further explain Table 5, with a focus on the MOTHEALTH factor and considering a positive change in the mean value of the specific factor (due to an increase in the importance attributed by parents to the physical and mental health of their students) within the range of up to +20%, it is evident that for any 1% increase in the attributed importance to the questionnaire parameters comprising the factor MOTHEALTH, the corresponding increase in the percentage of parents choosing a non-motorized travel mode will be 0.48% (Table 6). This is equivalent to an additional 4.8 out of 1000 parents opting for non-motorized school commuting. Conversely, for any 1% decrease in the attributed importance to the questionnaire parameters of the same factor, the percentage of parents choosing non-motorized travel modes will decrease by 0.11% (or 1.1 out of 1000 parents will shift from non-motorized to motorized commuting). Similar variations, in the range of −20% to +20%, in the values of the questionnaire parameters comprising the other factors lead to changes in the percentages of choosing non-motorized school commuting, as depicted in Table 6.

In fact, Table 6 represents one of the key scientific findings of this research, documenting the impact of changes in the beliefs, perceptions, and established habits of parents in urban commuting, as well as the perceived safety of the route and neighborhood between home and school, on the variation in the percentage of non-motorized transport mode usage. More specifically, from Table 6 it is evident that each 1% increase in the importance parents attribute to the questionnaire parameters of the MOTHEALTH factor (for the questionnaire parameters of each factor, see Table A2), which summarizes the parameters relevant to the physical and mental health of children that parents consider when choosing the transport mode of school commuting, leads to a 0.48% enhancement in the percentage of non-motorized mode usage. Similarly, every 1% increase in the parents' perceived sense of safety provided by sidewalks and the entire path the student follows between their residence and the school unit (factor ROUTESAF) augments the percentage of non-motorized travel mode usage by 0.37%. Conversely, a potential decrease of 1% in the parents' perceived safety of the route between their residence and the school unit would result in a reduction in children's non-motorized travel by −0.58%. A similar rationale can be followed for estimating the impacts of changes in the mean values of the other factors on the choice of the school travel mode choice and the differentiation between motorized and non-motorized modes.

**Table 6.** Impact of a 1% adjustment (decrease or increase) in the mean values of each factor on parental preference for non-motorized school transport modes.

| Labeled Factors (Latent Variables) | Change in the Percentage of Non-Motorized School Commuting Resulting from a 1% Decrease in the Mean Values of the Specific Factor | Change in the Percentage of Non-Motorized School Commuting Resulting from a 1% Increase in the Mean Values of the Specific Factor |
| --- | --- | --- |
| MOTHEALTH: Parameters of parental motivation to choose a transport mode related to the physical and mental health of a student | –0.11% [1] or 1.1 out of 1000 parents will shift from non-motorized to motorized modes | +0.48% or an additional 4.8 out of 1000 parents will opt for non-motorized modes |
| ATTBUS: Parents perception regarding the use of public buses | +0.33% or an additional 3.3 out of 1000 parents will opt for non-motorized modes | –0.16% or 1.6 out of 1000 parents will shift from non-motorized to motorized modes |
| ATTCAR: Parents perception regarding the use of private vehicles | +0.21% or an additional 2.1 out of 1000 parents will opt for non-motorized modes | –0.22% or 2.1 out of 1000 parents will shift from non-motorized to motorized modes |

**Table 6.** *Cont.*

| Labeled Factors (Latent Variables) | Change in the Percentage of Non-Motorized School Commuting Resulting from a 1% Decrease in the Mean Values of the Specific Factor | Change in the Percentage of Non-Motorized School Commuting Resulting from a 1% Increase in the Mean Values of the Specific Factor |
|---|---|---|
| ATTWALKBIK: Parents perception regarding the use of non-motorized transport modes (walking, bicycle) | −0.03% Marginal change not worthy of comment | +0.24% or an additional 2.4 out of 1000 parents will opt for non-motorized modes |
| MOTCAR: Parameters related to the usability of private vehicles | +0.13% or an additional 1.3 out of 1000 parents will opt for non-motorized modes | −0.13% or 1.3 out of 1000 parents will shift from non-motorized to motorized modes |
| NEIGBSAF: Parameters related to the sense of security provided by the neighborhood | −0.08% Marginal change not worthy of comment | +0.01% Marginal change not worthy of comment |
| ROUTESAF: Parameters related to the sense of safety provided by the sidewalks and the whole path the student follows | −0.58% or 5.8 out of 1000 parents will shift from non-motorized to motorized modes | +0.37% or an additional 3.7 out of 1000 parents will opt for non-motorized modes |

[1] All values apply to changes of factors' mean values within the range −20% to +20%. Outside of this range, the values differ due to non-linearities.

## 3. Discussion

In the forthcoming discussion section, we delve into three pivotal research questions that lie at the heart of our study, as stated clearly in Section 2.3.

The first research inquiry revolves around the capabilities of ML classification techniques, specifically their aptitude to leverage parental responses to diverse survey questions, which collectively form the observed variables. We aim to ascertain whether these techniques can effectively predict parental choices concerning school transportation modes, encompassing both motorized and non-motorized options. Furthermore, we seek to quantify the extent of accuracy achievable through these predictive models.

ML classification techniques offer powerful and highly effective means to predict parental choices when it comes to selecting between motorized and non-motorized school transportation, and this prediction is achieved with a remarkable level of precision, exemplified by impressive classifying performance scores such as an F1-score at 89% and an MCC at 80% (Table 3). The remarkable success of the classification model in this context can be primarily attributed to a confluence of critical factors.

First and foremost, the richness of the data source plays a pivotal role. By harnessing the information encapsulated in parents' responses to a diverse set of survey questions, the model can tap into a comprehensive array of factors that inherently influence parental decisions regarding school transportation. Furthermore, the art of feature engineering takes center stage. The meticulous selection and transformation of relevant features derived from the survey data are instrumental in unraveling intricate patterns and relationships within the dataset, enabling the model to make highly accurate predictions that align with the intricacies of real-world parental decision-making processes.

The complexity of the employed AI techniques is yet another contributing factor. These techniques can model complex interactions and non-linear relationships among the variables, resulting in a representation of decision-making processes that closely mirrors reality. In addition, the quality of the data is of paramount importance. Rigorous data preprocessing and cleaning measures ensure that the dataset is devoid of inconsistencies or errors, laying a solid foundation for the model's performance.

Lastly, the model's ability to generalize from the training data to make accurate predictions on new, unseen data is a testament to its robustness. With impressive classifying performance scores such as an F1-score at 89% and an MCC at 80% (Table 3), the model is clearly not overfitting to the training data but rather demonstrates a high level of pro-

ficiency in capturing the complexities of parental decision making in the realm of school transportation.

In conclusion, the application of AI classification techniques to predict parental choices in the context of school transportation is underpinned by meticulous data handling, strategic feature selection, model sophistication, and thorough evaluation. With impressive classifying performance scores such as an F1-score at 89% and an MCC at 80% (Table 3), the model exemplifies a remarkable accuracy in understanding the intricate dynamics of parental decision making.

The second research question delves into the influence wielded by two critical factors, NEIGBSAF and ROUTESAF, that lie within the purview (jurisdiction) of the state and municipal authorities. These entities have the power to effect change in the short and medium term through decision making for improvement and rehabilitation initiatives.

As proved in the previous section, an improvement across the route a student follows from their residence to the school unit and vice versa plays a crucial role in parents' decisions regarding the use of non-motorized transport modes. This also reflects the case where improvements in the environment surrounding the school unit are attempted; however, to a lesser degree. The design and implementation of infrastructure measures by the local authorities in full cooperation with the relevant stakeholders (school administrations, school committees, transport operators, micro-mobility providers, police, mobility planners, etc.) is essentially critical for a successful school mobility management, thus should not be ignored. The creation of infrastructure improvements along streets and within neighborhoods that ensure safer and more comfortable walking and bicycling and reduce injury risks should be at the top of local authorities' agendas. Examples of infrastructure improvements may include the rearrangement, extension, repair, or even reconstruction of sidewalks and cycling lanes; the introduction of speed reduction measures (traffic lights, zone 30 km/h, etc.); redesigning safer and more comfortable intersections near the school units with the use of specific elements (construction of safety islands, roundabouts, etc.); improvement of lighting along the streets; and the recreation/redevelopment of abandoned areas.

The last research question focuses on the examination of five factors, MOTHEALTH, ATTBUS, ATTCAR, ATTWALKBIKE, and MOTCAR, that encompass parental perceptions and beliefs. These factors are indirectly influenced by the state through long-term strategies such as targeted information provision, awareness-raising and marketing campaigns, organization of school travel plans, technological advancements such as gamification, etc.

School plays a strategic role in promoting the culture of sustainable mobility. Thus, school administrations should act as the driving force behind these initiatives, taking overactive roles in their planning and strongly supporting their contribution to behavioral change towards sustainable mobility. It is necessary, though, to provide teachers and other educators opportunities and incentives for acquiring new competencies related to school mobility needs and requirements, while it is an imperative need to share the acquired knowledge with students and parents by involving them in public presentations, events, and discussions relevant to school mobility.

## 4. Conclusions

This research endeavors to provide a valuable decision support tool for urban planners and policymakers, harnessing the power of state-of-the-art AI classifiers. Through this approach, we aim to facilitate the development of strategic initiatives and measures that promote equitable, safe, and sustainable school mobility systems.

The current work stands at the forefront of innovation, marking a pioneering exploration in the application of ML classification techniques to questionnaire datasets within the realm of school travel mode choice. To the best of our knowledge, this marks the inaugural instance of engaging such advanced methodologies in this specific domain. This groundbreaking research not only extends the boundaries of existing knowledge but also establishes a new paradigm for understanding and analyzing the factors influencing school travel mode decisions, thereby contributing significantly to the advancement of research in

this field. The utilization of ML in conjunction with questionnaire data lays the groundwork for novel insights and holds promise for enhancing the effectiveness of decision-making processes related to school commuting.

By delving into a comprehensive literature review, we have identified a multitude of factors influencing the school travel mode choice process, while also highlighting AI classification applications' effectiveness in forecasting human behavior traits.

After gaining a deeper understanding of the dynamics behind parental transportation choices, we can inform policies that enhance public transport access, optimize school environments, encourage active transport modes, raise awareness, and provide incentives, ultimately contributing to a more efficient and effective school transportation system. The synergy of AI classification techniques and a thorough exploration of these factors positions this study as a valuable resource for shaping the future of school transportation planning and urban mobility.

Our proposed implementation has been developed as a proof of concept for a decision support system for transportation infrastructure planners to enlighten them in understanding how they could achieve a desirable non-motorized school mode travel choice, perturbing mobility factors. In the short term, our approach demonstrated an actionable suggestion of which mobility factor variation would trigger a greater impact for improving the non-motorized school travel mode choice against the motorized school travel mode choice.

One of the limitations of the current work was identified as being the relatively small but still statistically acceptable sample of questionaries and the fact that they were gathered only from the city of Thessaloniki. In future work, our vision is targeted to selecting more data from more cities around the world, which will help us establish a robust framework able to generalize and extend to other case studies.

Moreover, in future work, more sophisticated AI methodologies could be investigated, such as deep learning techniques to augment classification ability, to help us comprehend the way parental beliefs influence the school travel mode choice. Thus, we encourage the community to dedicate more effort to the research and development of such AI synergizing frameworks and toolboxes, which will promote a more safe and sustainable school mobility ecosystem.

**Author Contributions:** Conceptualization, S.K., K.M.K., and G.B.; methodology, S.K., K.M.K., G.B., P.F.K., and D.S.; software, S.K., G.B., and P.F.K.; validation, S.K., K.M.K., G.B., P.F.K., and D.S.; formal analysis, S.K., K.M.K., G.B., P.F.K., and D.S.; investigation, S.K., K.M.K., G.B., P.F.K., and D.S.; resources, S.K. and K.M.K.; data curation, S.K. and K.M.K.; writing—original draft preparation, S.K., K.M.K., and G.B.; writing—review and editing, S.K., K.M.K., G.B., P.F.K., and D.S.; visualization, S.K. and G.B.; supervision, S.K., K.M.K., and G.B. All authors have read and agreed to the published version of the manuscript.

**Funding:** This research received no external funding.

**Institutional Review Board Statement:** Not applicable.

**Informed Consent Statement:** Not applicable.

**Data Availability Statement:** No new data were created or analyzed in this study. Data sharing is not applicable to this article.

**Conflicts of Interest:** The authors declare no conflicts of interest.

## Appendix A

Table A1 depicts the survey questionnaire. Further details regarding the structure, the composition, and the sample of the questionnaire survey are provided in [1,52,53].

**Table A1.** The questionnaire of the survey.

<table>
<tr><td colspan="6"><b>Research On School Transport</b><br>(The survey is addressed to parents of students aged 6–18 years)</td></tr>
<tr><td>Q1. Gender:</td><td>☐ Man</td><td>☐ Woman</td><td></td><td></td><td></td></tr>
<tr><td>Q2. Age category:</td><td>☐ 24–29</td><td>☐ 30–39</td><td>☐ 40–49 ☐</td><td>☐ 50–59</td><td>☐ 60–64</td></tr>
<tr><td>Q3. Family status:</td><td>☐ Married/in cohabitation</td><td>☐ Divorced</td><td></td><td></td><td></td></tr>
<tr><td>Q4. Educational level:</td><td>☐ Elementary School</td><td>☐ High School</td><td>☐ University Degree</td><td>☐ M.Sc./Ph.D. ☐</td><td></td></tr>
<tr><td>Q5. Work status:</td><td>☐ Full time</td><td>☐ Part time</td><td>☐ Unemployed</td><td></td><td></td></tr>
<tr><td></td><td>☐ Unpaid (household)</td><td>☐ University Student</td><td>☐ Retired</td><td></td><td></td></tr>
<tr><td>Q6. Driver license possession:</td><td>☐ At least one parent</td><td>☐ Both parents</td><td>☐ None</td><td></td><td></td></tr>
<tr><td>Q7. Number of private vehicles in the family:</td><td>☐ 0</td><td>☐ 1</td><td>☐ 2</td><td>☐ 3 or more</td><td></td></tr>
<tr><td>Q8. Student's age: ___ years old</td><td></td><td></td><td></td><td></td><td></td></tr>
<tr><td>Q9. Students' gender:</td><td>☐ Boy</td><td></td><td>☐ Girl</td><td></td><td></td></tr>
<tr><td>Q10. Please write down the student's school unit:<br><br>_______________________________</td><td></td><td></td><td></td><td></td><td></td></tr>
<tr><td>Q11. The school unit is located in:</td><td>☐ Urban environment</td><td>☐ Rural environment</td><td></td><td></td><td></td></tr>
<tr><td>Q12. What is the approximate distance of your residence to the school unit?</td><td>☐ up to 0.5 km</td><td>☐ 1.0–1.5 km</td><td>☐ 1.5–2.0 km</td><td>☐ 2.0–2.5 km</td><td>☐ 2.5–3.0 km</td></tr>
<tr><td>Q13. Has your child ever expressed the desire to travel to school by bicycle?</td><td>☐ Yes</td><td>☐No</td><td></td><td></td><td></td></tr>
<tr><td>Q14. I would allow my child to walk or cycle to and from school alone without any parent accompanying them:</td><td>☐ Totally agree</td><td>☐ Agree</td><td>☐ Neither agree or disagree</td><td>☐ Disagree</td><td>☐ Totally disagree</td></tr>
<tr><td>Q15. What transport mode does your child uses from residence to school unit and from school unit to residence. If there is a difference in the transport modes due to summer or winter, please note the predominant one:</td><td>Q15-1. Residence → School unit</td><td>Q15-2. School unit → Residence</td><td></td><td></td><td></td></tr>
<tr><td>Walking alone</td><td>☐</td><td>☐</td><td></td><td></td><td></td></tr>
<tr><td>Walking, with friends</td><td>☐</td><td>☐</td><td></td><td></td><td></td></tr>
<tr><td>Walking, accompanied by the respondent</td><td>☐</td><td>☐</td><td></td><td></td><td></td></tr>
<tr><td>Walking, accompanied by an adult</td><td>☐</td><td>☐</td><td></td><td></td><td></td></tr>
<tr><td>Urban public bus</td><td>☐</td><td>☐</td><td></td><td></td><td></td></tr>
<tr><td>School bus, paid by school</td><td>☐</td><td>☐</td><td></td><td></td><td></td></tr>
<tr><td>I drive him/her to school</td><td>☐</td><td>☐</td><td></td><td></td><td></td></tr>
<tr><td>A friend of mine drives him/her to school</td><td>☐</td><td>☐</td><td></td><td></td><td></td></tr>
<tr><td>Taxi</td><td>☐</td><td>☐</td><td></td><td></td><td></td></tr>
<tr><td>Bicycle</td><td>☐</td><td>☐</td><td></td><td></td><td></td></tr>
<tr><td colspan="6">Q16. What transport mode would you prefer for your child to use from residence to school unit and from school unit to residence:</td></tr>
<tr><td>Walking alone</td><td>☐</td><td></td><td></td><td></td><td></td></tr>
<tr><td>Walking, with friends</td><td>☐</td><td></td><td></td><td></td><td></td></tr>
<tr><td>Walking, accompanied by the respondent</td><td>☐</td><td></td><td></td><td></td><td></td></tr>
<tr><td>Walking, accompanied by an adult</td><td>☐</td><td></td><td></td><td></td><td></td></tr>
<tr><td>Urban public bus</td><td>☐</td><td></td><td></td><td></td><td></td></tr>
<tr><td>School bus, paid by school</td><td>☐</td><td></td><td></td><td></td><td></td></tr>
<tr><td>I drive him/her to school</td><td>☐</td><td></td><td></td><td></td><td></td></tr>
<tr><td>A friend of mine drives him/her to school</td><td>☐</td><td></td><td></td><td></td><td></td></tr>
</table>

**Table A1.** *Cont.*

**Research On School Transport**
(The survey is addressed to parents of students aged 6–18 years)

| | | | | | |
|---|---|---|---|---|---|
| Taxi | ☐ | | | | |
| Bicycle | ☐ | | | | |

Q17. According to your opinion which is the level of importance for each of the following factors that prompt you to select the specific transport mode?

| | Not important at all | | | | Absolutely important |
|---|---|---|---|---|---|
| | [1] | [2] | [3] | [4] | [5] |
| Q17-1. Student gender | ☐ | ☐ | ☐ | ☐ | ☐ |
| Q17-2. Student age | ☐ | ☐ | ☐ | ☐ | ☐ |
| Q17-3. There is someone to assist me with school transportation | ☐ | ☐ | ☐ | ☐ | ☐ |
| Q17-4. Working hours | ☐ | ☐ | ☐ | ☐ | ☐ |
| Q17-5. Personal/family income | ☐ | ☐ | ☐ | ☐ | ☐ |
| Q17-6. Driving license possession | ☐ | ☐ | ☐ | ☐ | ☐ |
| Q17-7. Car ownership | ☐ | ☐ | ☐ | ☐ | ☐ |
| Q17-8. Limitations on parking | ☐ | ☐ | ☐ | ☐ | ☐ |
| Q17-9. Distance from school | ☐ | ☐ | ☐ | ☐ | ☐ |
| Q17-10. Time spent on trip | ☐ | ☐ | ☐ | ☐ | ☐ |
| Q17-11. Trip cost | ☐ | ☐ | ☐ | ☐ | ☐ |
| Q17-12. Student's comfort | ☐ | ☐ | ☐ | ☐ | ☐ |
| Q17-13. Student's safety | ☐ | ☐ | ☐ | ☐ | ☐ |
| Q17-14. Environmental sensitivities | ☐ | ☐ | ☐ | ☐ | ☐ |
| Q17-15. Student's health | ☐ | ☐ | ☐ | ☐ | ☐ |
| Q17-16. School luggage weight | ☐ | ☐ | ☐ | ☐ | ☐ |
| Q17-17. Socializing with friends | ☐ | ☐ | ☐ | ☐ | ☐ |
| Q17-18. Spend quality time with child | ☐ | ☐ | ☐ | ☐ | ☐ |

Q18. In what degree to you agree or disagree with the below statements regarding the student's school trip?

| | Strongly disagree | | | | Strongly agree |
|---|---|---|---|---|---|
| | [1] | [2] | [3] | [4] | [5] |
| Q18-1. Traffic conditions are not dangerous | ☐ | ☐ | ☐ | ☐ | ☐ |
| Q18-2. I believe there are safe intersections | ☐ | ☐ | ☐ | ☐ | ☐ |
| Q18-3. I find it unlikely my child to be abducted or injured by a stranger | ☐ | ☐ | ☐ | ☐ | ☐ |
| Q18-4. I find it unlikely my child to be harassed by others | ☐ | ☐ | ☐ | ☐ | ☐ |
| Q18-5. The route from residence to school is safe | ☐ | ☐ | ☐ | ☐ | ☐ |
| Q18-6. There are sidewalks of adequate width | ☐ | ☐ | ☐ | ☐ | ☐ |
| Q18-7. Sidewalks are clean | ☐ | ☐ | ☐ | ☐ | ☐ |
| Q18-8. Sidewalks are separated by traffic with trees | ☐ | ☐ | ☐ | ☐ | ☐ |
| Q18-9. There are no obstacles on sidewalks (rubbish bins, parked cars, etc.) | ☐ | ☐ | ☐ | ☐ | ☐ |
| Q18-10. Residents in the neighborhood are in good condition | ☐ | ☐ | ☐ | ☐ | ☐ |
| Q18-11. There are no vandalism traces in our neighborhood | ☐ | ☐ | ☐ | ☐ | ☐ |
| Q18-12. There is adequate lighting on the route from residence to school unit | ☐ | ☐ | ☐ | ☐ | ☐ |
| Q18-13. The existing infrastructures cannot protect a cyclist | ☐ | ☐ | ☐ | ☐ | ☐ |

Q19. In what degree to you agree or disagree with the below statements regarding the student's school trip?

| | Strongly disagree | | | | Strongly agree |
|---|---|---|---|---|---|
| | [1] | [2] | [3] | [4] | [5] |
| Q19-1. Travelling to school on foot/bike is a good way my child to become familiar with the neighborhood | ☐ | ☐ | ☐ | ☐ | ☐ |
| Q19-2. I would like my child to travel to school on foot or by bike under the appropriate circumstances | ☐ | ☐ | ☐ | ☐ | ☐ |

**Table A1.** *Cont.*

| **Research On School Transport** (The survey is addressed to parents of students aged 6–18 years) | | | | |
|---|---|---|---|---|
| Q19-3. Travelling to school on foot or by bike is a way to increase my child's physical activity | ☐ | ☐ | ☐ | ☐ | ☐ |
| Q19-4. Driving my child to school may lead to car use addiction | ☐ | ☐ | ☐ | ☐ | ☐ |
| Q19-5. Driving to school contributes to driving congestion | ☐ | ☐ | ☐ | ☐ | ☐ |
| Q19-6. Driving is more comfortable than walking/cycling | ☐ | ☐ | ☐ | ☐ | ☐ |
| Q19-7. I like driving within the city | ☐ | ☐ | ☐ | ☐ | ☐ |
| Q19-8. Owning a car makes my life more comfortable | ☐ | ☐ | ☐ | ☐ | ☐ |
| Q19-9. I use my car even for short distances | ☐ | ☐ | ☐ | ☐ | ☐ |
| Q19-10. Car ownership is a prestige symbol | ☐ | ☐ | ☐ | ☐ | ☐ |
| Q19-11. Traffic congestion doesn't bother me | ☐ | ☐ | ☐ | ☐ | ☐ |
| Q19-12. I like to use the urban bus for travelling within the city | ☐ | ☐ | ☐ | ☐ | ☐ |
| Q19-13. The urban bus is a very reliable transport mode | ☐ | ☐ | ☐ | ☐ | ☐ |
| Q19-14. I am satisfied with the comfort of the urban bus | ☐ | ☐ | ☐ | ☐ | ☐ |
| Q19-15. I am satisfied with the time consistency of the urban bus services | ☐ | ☐ | ☐ | ☐ | ☐ |

**Appendix B**

Table A2 presents grouped parameters (observed variables) derived from both exploratory (EFA) and confirmatory factor analysis (CFA), constituting labeled factors (latent variables). EFA identified underlying factors by exploring the data structure, while CFA validated factor structures predetermined by EFA. EFA explored and CFA confirmed predefined relationships between the observed and latent variables in the survey data, contributing to a nuanced understanding of latent constructs.

The labeling of the extracted factors was conducted with meticulous consideration of the semantic content inherent in the questionnaire parameters (observed variables, Appendix B, 2nd column of Table A2) encapsulated within each respective factor. In the formulation of the acronyms for the extracted factors, the prefix "MOT" (motivation) was selected for factors incorporating parameters related to motivations regarding the utilization of a transport mode. Additionally, the prefix "ATT" (attitudes) was chosen for factors comprising questionnaire parameters associated with behavioral aspects and shaped perceptions. Notably, the suffix "SAF" (safety) was employed for factors encompassing variables pertaining to safety within the school and residence neighborhoods (NEIGB) or along the school route (ROUTE).

Detailed analysis of the EFA and CFA can be found in [1,52,53].

**Table A2.** Labelling of the factors derived from the exploratory and confirmatory factor analysis.

| Questionnaire Parameters (Observed Variables) | | | Labeled Factors (Latent Variables) | |
|---|---|---|---|---|
| No | Description | Mean Value (Standard Deviation) in 5-Point Likert Scale | Acronym and Description (in Parenthesis the Reliability Coefficient Cronbach's Alpha) | Mean Value in 5-Point Likert Scale |
| Q17-2 | Importance of student age | 4.06 (1.13) | MOTMODE (0.88): Objective parameters of parental motivation to choose a transport mode | 3.84 (1: Not important at all, 5: Absolutely important) |
| Q17-3 | There is someone to help | 3.45 (1.31) | | |
| Q17-4 | Working hours | 3.74 (1.26) | | |
| Q17-9 | Distance residence → school | 3.96 (1.14) | | |
| Q17-10 | Travel time residence → school | 3.75 (1.16) | | |
| Q17-12 | Convenience | 3.69 (1.17) | | |
| Q17-13 | Student's safety | 4.21 (1.18) | | |
| Q17-14 | Environmental sensitivities | 3.12 (1.18) | MOTHEALTH (0.88): Parameters of parental motivation to choose a transport mode related to the physical and mental health of a student | 3.39 (1: Not important at all, 5: Absolutely important) |
| Q17-15 | Student's health | 3.57 (1.23) | | |
| Q17-16 | School luggage weight | 3.62 (1.24) | | |
| Q17-17 | Socialization with friends | 3.54 (1.20) | | |
| Q17-18 | Quality time between parent and child | 3.09 (1.20) | | |
| Q19-12 | I like travelling by urban bus within the city | 2.64 (1.16) | ATTBUS (0.82): Parents perception regarding the use of public buses | 2.26 (1: Strongly disagree, 5: Strongly agree) |
| Q19-13 | Urban bus is a reliable transport mode | 2.37 (1.11) | | |
| Q19-14 | Satisfied with the comfort of urban bus services | 2.01 (1.08) | | |
| Q19-15 | Satisfied with time reliability with urban bus services | 2.00 (1.12) | | |
| Q15-1, 2 | Transport mode, residence → school and vice-versa | Not applicable | MODE (0.86): Selected and preferred school travel mode choice | Not applicable |
| Q16 | Preferable transport mode | | | |
| Q19-6 | Driving is more comfortable than walking or bicycling | 2.81 (1.12) | ATTCAR (0.87): Parents perception regarding the use of private vehicles | 3.06 (1: Strongly disagree, 5: Strongly agree) |
| Q19-7 | I like driving within the city | 2.83 (1.16) | | |
| Q19-8 | Owing a car makes my life comfortable | 3.76 (0.93) | | |
| Q19-9 | I use my car for all trips within the city | 2.85 (1.18) | | |

**Table A2.** *Cont.*

| No | Description | Mean Value (Standard Deviation) in 5-Point Likert Scale | Acronym and Description (in Parenthesis the Reliability Coefficient Cronbach's Alpha) | Mean Value in 5-Point Likert Scale |
|---|---|---|---|---|
| | **Questionnaire Parameters (Observed Variables)** | | **Labeled Factors (Latent Variables)** | |
| Q19-1 | Walking or bicycling to school is a good way my child be familiar with environment | 3.89 (0.79) | ATTWALKBIK (0.72): Parents perception regarding the use of non-motorized transport modes (walking, bicycle, etc.) | 4.14 (1: Strongly disagree, 5: Strongly agree) |
| Q19-2 | I would prefer my child walk or drive to school under different circumstances | 4.27 (0.77) | | |
| Q19-3 | Walking or cycling to school increases students' physical activity | 4.25 (0.70) | | |
| Q17-6 | Driving license possession | 3.04 (1.33) | MOTCAR (0.95): Parameters related to the usability of private vehicles | 3.07 (1: Not important at all, 5: Absolutely important) |
| Q17-7 | Car ownership | 3.12 (1.34) | | |
| Q17-8 | There are no parking limitations outside my residence or the school unit | 3.06 (1.30) | | |
| Q18-5 | The neighborhood the student travels is safe | 2.88 (1.08) | NEIGBSAF (0.76): Parameters related to the sense of security provided by the neighborhood | 3.29 (1: Strongly disagree, 5: Strongly agree) |
| Q18-10 | Residences of neighborhood are in good conditions | 3.35 (1.12) | | |
| Q18-11 | There are no vandalism traces in the neighborhood | 3.23 (1.05) | | |
| Q18-1 | Traffic conditions are not dangerous for the students | 2.18 (1.10) | ROUTESAF (0.91): Parameters related to the sense of safety provided by the sidewalks and the whole path the student follows | 2.30 (1: Strongly disagree, 5: Strongly agree) |
| Q18-2 | Crossings are safe | 2.27 (1.15) | | |
| Q18-3 | It's unlikely for my child to be injured or abducted by | 2.27(1.11) | | |
| Q18-4 | It's unlikely for my child to be harassed by others | 2.23 (1.07) | | |
| Q18-6 | Sidewalks have sufficient width | 2.25 (1.17) | | |
| Q18-7 | Sidewalks are clean | 2.32 (1.15) | | |
| Q18-8 | Sidewalks are separated from traffic with trees | 2.07 (1.10) | | |
| Q18-9 | There are no obstacles in the sidewalks | 2.15 (1.14) | | |
| Q18-12 | There is adequate lighting in the school trip route | 2.95 (1.17) | | |

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
