# Peer review of "Assessing Impact Factors That Affect School Mobility Utilizing a Machine Learning Approach"

_sustainability, doi:10.3390/su16020588_

Round 1

Reviewer 1 Report

Comments and Suggestions for Authors

Full Title: Assessing impact factors that affect school mobility utilizing a novel Artificial Intelligence approach

Objective: The paper investigates the characteristics, constraints, and importance of predictions related to school travel mode choices, particularly focusing on the city of Thessaloniki, Greece. It conducts a literature review on traditional models and explores recent Artificial Intelligence (AI) approaches in mode choice modeling. Through a questionnaire survey, the study identifies objective parameters of parental motivation for selecting specific transport modes and analyzes parents’ perceptions of motorized and non-motorized options, as well as considerations about the surrounding environment. The paper uses Exploratory and Confirmatory Factor Analysis to group survey questions, followed by the development of a Machine Learning (ML) classification model trained on parental responses. This model predicts parental decisions between motorized and non-motorized school travel modes, allowing the study to assess the impact of various factors on school travel choices. The ultimate goal is to evaluate and promote factors contributing to a more equitable, safer, and sustainable non-motorized school mobility system.

There are some issues that need to be addressed:

1.      The literature review section should stand alone. Include a table summarizing the works reviewed in this section. Additionally, establish a clear connection between the theoretical background and the research design or model. Specifically, pinpoint additional literature that is more pertinent to the predictive factors.

  1. In the data collection section, provide details regarding the profile of the respondents, including information on age, gender, and other relevant demographic factors.
  2. Kindly present the outcomes of the Exploratory Factor Analysis (EFA), including the Pattern Matrix or Rotated Component Matrix. Additionally, report Commonality values, offering insights into the proportion of variance explained by the factors. Furnish Descriptive Statistics such as Mean, Standard Deviation (S.D.), Skewness, and Kurtosis for each subdimension to provide a comprehensive understanding of the data distribution. Furthermore, include Reliability coefficients, such as Cronbach’s alpha, for each subdimension, elucidating the internal consistency of the identified factors.
  3. Kindly present the results of the Confirmatory Factor Analysis (CFA) conducted within the study. Assess both convergent and discriminant validity, and provide the Heterotrait-Monotrait (HTMT) results. Report the model fit indices, including x2, RMSEA, AGFI, GFI, NFI, CFI, PGFI, and other relevant metrics, to evaluate the adequacy of the measurement model.

5.      Employing the same data set for both Exploratory Factor Analysis (EFA) and Confirmatory Factor Analysis (CFA) is not recommended due to concerns regarding common method variance (Campbell, 1976). Additionally, Veerson and Gerbing (1991) propose that when an item is removed or a new item is introduced to the scale, the updated scale should be applied to a distinct independent sample. Following this recommendation, researchers ought to gather data from a second sample group, utilizing the remaining items derived from the exploratory factor analysis. Subsequently, Confirmatory Factor Analysis (CFA) should be undertaken to validate the factor structure of the scale using the second sample.

6.      In the ML classification model, enumerate and elucidate the features, encompassing both dependent and independent variables, that constitute the school mobility prediction model. In the Method section, furnish a comprehensive description of the dataset employed, offering insights into its composition, structure, and relevant characteristics to enhance the understanding of the applied methodology.

7.      Providing the Machine Learning (ML) classification model in a figure (TIFF images with minimum 300 dpi) showing the relationships between study variables would be better.

8.      How many predictors were specified in the ML classification model, and are they continuous or categorical variables? Additionally, how were the classes of the dependent variable defined?

9.      Likewise, provide descriptive statistics (means, SD) about the features or attributes (independent variables) of the model.

10.  Provide classifiers’ performance indicators (CCI, TP Rate, FP Rate, Precision, Recall, F-Measure, MCC, ROC Area) in a table.

11.  Provide Mean Absolute Error (MAE), Root Mean Squared Error (RMSE), and Matthew’s Correlation Coefficient (MCC) values for each model. RMSE is a quadratic metric that measures the size of the error that is often used to find the distance between the predicted values and the observed values of the estimator. Therefore, it is important to report RMSE to properly assess the performance of the model.

12.  Please conduct a sensitivity analysis to identify the most important input factors.

13.  Present a benchmarking table that compares your study's findings (classification accuracy or performance) with those of previous works.

Comments on the Quality of English Language

Proof reading is required.

Author Response

Reviewer 1

Νο

Comments and suggestions to authors - Reviewer #1

Authors’ reply to the Review Report #1

There are some issues that need to be addressed:

First and foremost, we would like to express our warm thanks to the reviewer for the time and effort devoted to our work and for the valuable observations and suggestions aimed at improving the paper.

1

The literature review section should stand alone. Include a table summarizing the works reviewed in this section. Additionally, establish a clear connection between the theoretical background and the research design or model. Specifically, pinpoint additional literature that is more pertinent to the predictive factors.

The literature review follows a twofold dimension as it has been already described in the relevant section (1.3). Therefore, and according to this specific reviewer’s comment, the authors proceeded in the inclusion of two different Tables (Table 1 and Table 2), summarizing the works reviewed.

Table 1 includes research works according to similar key findings (mainly defines the critical factors that have been found to affect the school travel mode choice). Table 2 summarizes research works regarding the deployment of AI approaches in the transport domain trying to expand and focus on school transportation.

The theoretical background and the research design is well explained within the text as it is highlighted that the authors through the literature review defined the critical parameters affecting the school travel mode choice. This helped them in setting up the Questionnaire Survey (basic tool for the primary research needs - data collection).

Additionally, it is further explained that due to limited research work (theoretical background) in testing AI approaches for the school travel mode choice prediction, authors move a step forward and develop a state-of-the-art AI approach to analyze behavioral patterns related to school mobility. A further literature review, added one more citation found to be relevant with AI applications in the school travel mode choice process.

Text amendments are highlighted in lines 112, table 1 at page 4, table 2 at pages 6 and 7. 

2

In the data collection section, provide details regarding the profile of the respondents, including information on age, gender, and other relevant demographic factors.

A brief description regarding the respondents’ profile has been added in the form of text in the relevant section (2.1), summarizing the Questionnaire Survey’s first part analysis results.

Text amendments are highlighted in lines 254 to 260.

3

Kindly present the outcomes of the Exploratory Factor Analysis (EFA), including the Pattern Matrix or Rotated Component Matrix. Additionally, report Commonality values, offering insights into the proportion of variance explained by the factors. Furnish Descriptive Statistics such as Mean, Standard Deviation (S.D.), Skewness, and Kurtosis for each subdimension to provide a comprehensive understanding of the data distribution. Furthermore, include Reliability coefficients, such as Cronbach’s alpha, for each subdimension, elucidating the internal consistency of the identified factors.

The reviewer is correct in their observation, as the present work does not delve into the analysis of Exploratory Factor Analysis (EFA) and Confirmatory Factor Analysis (CFA).

Nevertheless, as explicitly indicated in the text of this paper (lines 283 to 286), the analytical description and evaluation of the processes of EFA and CFA, along with the calculation of appropriate metrics, indices, and matrices, have already been published by the authors in prior researches [1, 51, 52] and are not reiterated in the present, in order to avoid gratuitous redundancies.

Despite this, for each factor, the reliability coefficient Cronbach's alpha was added to the 4th column of Table B1, as it did not increase the size of the study.

Additionally, in the revised manuscript, the percentage (%) of the overall variance in questionnaire parameters that is explained by the derived factors has been added (line 288).

Text amendments are highlighted in lines 283 to 286 and line 287.

4

Kindly present the results of the Confirmatory Factor Analysis (CFA) conducted within the study. Assess both convergent and discriminant validity, and provide the Heterotrait-Monotrait (HTMT) results. Report the model fit indices, including x2, RMSEA, AGFI, GFI, NFI, CFI, PGFI, and other relevant metrics, to evaluate the adequacy of the measurement model.

5

Employing the same data set for both Exploratory Factor Analysis (EFA) and Confirmatory Factor Analysis (CFA) is not recommended due to concerns regarding common method variance (Campbell, 1976). Additionally, Veerson and Gerbing (1991) propose that when an item is removed or a new item is introduced to the scale, the updated scale should be applied to a distinct independent sample. Following this recommendation, researchers ought to gather data from a second sample group, utilizing the remaining items derived from the exploratory factor analysis. Subsequently, Confirmatory Factor Analysis (CFA) should be undertaken to validate the factor structure of the scale using the second sample.

6

In the ML classification model, enumerate and elucidate the features, encompassing both dependent and independent variables, that constitute the school mobility prediction model. In the Method section, furnish a comprehensive description of the dataset employed, offering insights into its composition, structure, and relevant characteristics to enhance the understanding of the applied methodology.

In the revised version of the paper, Figure 3 has been included to schematically depict the ML classifier. This classifier is trained on parents' responses to a questionnaire survey, enabling it to predict factors associated with the decision-making process concerning school travel mode choice.

Also, in Figure 4 Pearson correlation coefficient matrix is expressed as a heatmap. The Pearson correlation coefficient can describe the linear correlation between two features.

Text amendments are highlighted in lines 374 to 378 and addendum of Figure 3 and 4.

7

Providing the Machine Learning (ML) classification model in a figure (TIFF images with minimum 300 dpi) showing the relationships between study variables would be better.

8

How many predictors were specified in the ML classification model, and are they continuous or categorical variables? Additionally, how were the classes of the dependent variable defined?

Since,

i) the individual rating items with numerical response formats at least five categories in length,

ii) the intervals between the (at least) five categories are assumed to be equal,

the 38 observed variables (questionnaire items) of Figure 3 may be treated as continuous data.

Concerning the choice of classes for the dependent variable, distinguishing between motorized and non-motorized transport, was based on the fact that non-motorized transport (NMT) often plays a crucial role in promoting clean urban transport. Non-motorized transportation, also known as active transportation and human-powered transportation, encompasses activities such as walking and bicycling, as well as variations like small-wheeled transport. These modes serve both recreational and transportation purposes and are particularly important for short trips up to 7 km (a distance that includes the majority of school commutes), which constitute a significant share of trips in urban areas. NMT can be encouraged through a policy package comprising investments in facilities and infrastructure for walking and bicycling, awareness campaigns, and disincentives for the use of motorized private vehicles, some of which are components included in the factor analysis conducted in this study.

No text amendment for this reviewer comment, only response stated above.

9

Likewise, provide descriptive statistics (means, SD) about the features or attributes (independent variables) of the model.

The mean values of 38 independent (observed) variables were already provided in Table B.1 (Appendix B). Subsequent to the reviewer's accurate suggestion, in the revised form of paper the standard deviations have also been incorporated into the same Table B.1.

Text amendments with addendum of Table B1.

10

Provide classifiers’ performance indicators (CCI, TP Rate, FP Rate, Precision, Recall, F-Measure, MCC, ROC Area) in a table.

In the revised version, we are providing Table 3. Classifiers performance, where we are presenting all ML models investigated and their performance, regarding the metrics requested. Also in line 439-462, Table 3 is annotated.

Text amendments are highlighted in lines 433 to 456 and addendum of Table 3.

11

Provide Mean Absolute Error (MAE), Root Mean Squared Error (RMSE), and Matthew’s Correlation Coefficient (MCC) values for each model. RMSE is a quadratic metric that measures the size of the error that is often used to find the distance between the predicted values and the observed values of the estimator. Therefore, it is important to report RMSE to properly assess the performance of the model.

Suggestion for MCC is embodied in revised version, as stated above. While MAE and RMSE are indeed common metrics in regression or forecasting models, they may not be suitable for evaluating the performance of a classification model. MAE and RMSE are metrics tailored to measure the accuracy of predicted numerical values against actual values in regression scenarios. In contrast, classification tasks involve assessing the correctness of assigning instances to predefined categories. As such, employing MAE and RMSE in a classification context may lead to misinterpretation and inappropriate evaluation of the model's performance.

No text amendment for this reviewer comment, only response stated above.

12

Please conduct a sensitivity analysis to identify the most important input factors.

One of the main research objectives was to identify factors with the greatest influence on the decision between motorized and non-motorized school travel mode choices, engaging AI techniques.  Leveraging the developed ML classification model, we assessed the impact of various factors on school travel mode choice, by selectively modifying parents' responses and evaluating the resulting change in the initial ratio of school travel mode choice between motorized and non-motorized school travel mode choice. Stated differently, our approach measured what would have happened to motorized and non-motorized school travels mode choice while modifying certain factors, and thus estimated the impact of each factor in motorized and non-motorized school travels mode choice. In that context, our modeling process estimated the impact of input factors’ changing on the outcomes of decision between motorized and non-motorized school travel mode choices, which encapsulates the fundamental essence of sensitivity analysis. Under that perspective authors highlight that sensitivity analysis is already performed, as a main research objective.

No text amendment for this reviewer comment, only response stated above.

13

Present a benchmarking table that compares your study's findings (classification accuracy or performance) with those of previous works.

This perspective was investigated during research. Still, there were no previous works related to school mode transportation choice implemented with ML techniques that use qualitative (attitudes, perception beliefs) to reviewers’ knowledge. Thus, benchmarking was not applicable.

As already stated at line 87-95, the current research signifies a groundbreaking and innovative contribution as it marks the inaugural application of ML classification techniques to a questionnaire dataset within the realm of school travel mode choice. To the best of our knowledge, this study stands as the first of its kind to leverage advanced ML methodologies in examining and categorizing qualitative and quantitative data related to the selection of school travel modes. This approach not only extends the boundaries of current research practices but elicits novel viewpoints for understanding the intricate dynamics influencing the decision-making process in the context of school transportation choices line 107-110

Text amendments are highlighted in line 89.

14

Comments on the Quality of English Language:

Proof reading is required.

Comment accepted. The proofreading suggested by the reviewer has been completed.

No text amendment for this reviewer comment, only response stated above.

Reviewer 2 Report

Comments and Suggestions for Authors

The article is very interesting and very well written, with a 'hot' topic and denotes the experience of its authors in this interdisciplinary research. I have read it with great pleasure and interest.

Hence my comments would normally be only to praise the outcome of this research. I particularly appreciate the acknowledgement of the limitations of the study, which considers only 496 responses although we do not know if this number is on a representative sample.

However there are a few issues that may deserve improvement:

1. In the literature review, e.g. (44), (45), (47) it is not clear whether the authors reached these conclusions after studying the responses for a relevant number of respondents. How many surveys were reviewed? What are the limitations of these articles? Thus the research can be better contextualised.

2. There is no Results section, only a Discussion section. Consequently, Figure 4, Tables 3,4,5 are explained very briefly. So I suggest explaining them in the text in more detail, taking into account that the paper is likely to be read also by non-specialists.

Good luck!

Author Response

Reviewer 2

Νο

Comments and suggestions to authors - Reviewer #2

Authors’ replies to the Review Report #2

1

The article is very interesting and very well written, with a 'hot' topic and denotes the experience of its authors in this interdisciplinary research. I have read it with great pleasure and interest.

Hence my comments would normally be only to praise the outcome of this research. I particularly appreciate the acknowledgement of the limitations of the study, which considers only 496 responses although we do not know if this number is on a representative sample.

However, there are a few issues that may deserve improvement:

First and foremost, we would like to express our warm thanks to the reviewer for the time and effort devoted to our work and for the valuable observations and suggestions aimed at improving the paper.

2

In the literature review, e.g. (44), (45), (47) it is not clear whether the authors reached these conclusions after studying the responses for a relevant number of respondents. How many surveys were reviewed? What are the limitations of these articles? Thus the research can be better contextualized.

Literature overview (44-49) is revised according to reviewer’s guidelines.

As already stated in manuscript in line 208-209 “The literature review highlighted a scarcity of research on school travel mode choice models, particularly in the realm of AI applications.”, the significant absence of relevant work was one of the main triggering factors, that led us to identify the need to further research this topic. Lack in extended research may be a barrier in generalization of such research in other case studies with different qualitative and quantitative characteristics. On the other hand, this is extremely challenging for researchers who want to pioneer and create new perspectives and innovative approaches in school mode travel choice and sustainable transportation.

Text amendments are highlighted in lines between 165 and 199.

3

There is no Results section, only a Discussion section. Consequently, Figure 4, Tables 3,4,5 are explained very briefly. So I suggest explaining them in the text in more detail, taking into account that the paper is likely to be read also by non-specialists.

The reviewer's comment is significant, and we acknowledge the challenges faced by a non-specialist in comprehending issues that simultaneously incorporate artificial intelligence and factor analysis. In an effort to make the findings of the study more understandable, the third paragraph of section 2.6 has been reformulated, and an additional paragraph (the last one in section 2.6) has been appended. providing a detailed explanation of the findings of Table 6, which is crucial for the study.

Text amendments are highlighted in lines 555 to 572.

Reviewer 3 Report

Comments and Suggestions for Authors

1.      The purpose of this article is to investigate the factors that affect students’ transport mode by Artificial Intelligence approach. Some critical questions listed below need to be identified before further consideration for publication.

2.      What and how can Artificial Intelligence do for this research?

3.      How to define the name of factors by Factor Analysis?

4.      What is result of factor analysis?

Author Response

Reviewer 3

No

Comments and suggestions to authors - Reviewer #3

Authors’ reply to the Review Report #3

The purpose of this article is to investigate the factors that affect students’ transport mode by Artificial Intelligence approach. Some critical questions listed below need to be identified before further consideration for publication.

First and foremost, we would like to express our warm thanks to the reviewer for the time and effort devoted to our work and for the valuable observations and suggestions aimed at improving the paper.

1

What and how can Artificial Intelligence do for this research?

Artificial Intelligence, specifically in the form of Machine Learning, serves as a powerful tool in this study, enabling the analysis and prediction of complex decision-making processes related to school travel mode choices based on a variety of parental considerations.

To assess the impact of various factors on school travel mode choices, authors selectively modify parents' responses. Leveraging the developed ML classification model, the study predicts the resulting change in the initial ratio of school commuting modes. This process allows identification of the most influential factors contributing to the establishment of an equitable, safer, and more sustainable non-motorized school mobility system.

No text amendment for this reviewer comment, only response stated above.

2

How to define the name of factors by Factor Analysis?

In pursuit of enhancing the comprehension of the research process, we incorporated the following text into the manuscript (lines 830 to 839), Appendix B.

This approach to factor naming ensures a nuanced representation of the latent constructs, aligning with both the measured parameters and the overarching conceptual framework of the research.

Text amendments are highlighted in lines 830 to 839.

3

What is result of factor analysis?

The (Exploratory and Confirmatory) Factor Analysis relevant to this research has been thoroughly examined and detailed in previous publications by the authors, with proper references provided in the text and the corresponding bibliography (references [1], [51]). The extensive nature of both Exploratory and Confirmatory Factor Analysis, which would significantly increase the length of the current paper (by at least 3, maybe 4 pages), and especially its prior publication in authors' earlier research manuscripts, influenced the decision to exclude the analytical processes of Exploratory and Confirmatory Factor Analysis in this paper. Only the final outcomes of the factor analysis are presented in Appendix B, Table B1

Text amendments are highlighted in lines 817 to 826.

Round 2

Reviewer 1 Report

Comments and Suggestions for Authors

The authors have addressed the majority of my comments effectively, yet a few additional concerns persist. Firstly, the present study utilizes Machine Learning (ML) classification algorithms to predict parents’ school travel mode choices (i.e., motorized and non-motorized modes) based on responses to a diverse array of survey questions encompassing factors such as working hours, convenience, car ownership, and students’ safety. The title, “Assessing impact factors that affect school mobility utilizing a novel Artificial Intelligence approach,” needs revision as it fails to accurately convey the study’s purpose. Since using ML classification algorithms as an AI approach is a standard technique for prediction, as evidenced in Table 2, the term “novel Artificial Intelligence approach” should be adjusted to "Utilizing Machine Learning approach."

Secondly, the authors mention that the processes of Exploratory Factor Analysis (EFA) and Confirmatory Factor Analysis (CFA) have been extensively detailed in prior research [1, 51, 52] to avoid unnecessary redundancy. However, the abstract suggests that "Survey questions are grouped into labeled factors using Exploratory and Confirmatory Factor Analysis." This implies that both EFA and CFA were indeed employed to identify and confirm factors in the present study. In light of this, it is imperative that the authors either present the results of the EFA and CFA in the current study or revise this line for clarity.

Given that new data from parents were collected, it is crucial to explicitly state whether CFA was conducted in the current study to confirm the factor structure for the gathered data. The inclusion of CFA results, if performed, will enhance the transparency of the analytical approach and provide readers with a comprehensive understanding of how factors were identified and validated in the specific context of the study.

Author Response

First and foremost, we would like to express our warm thanks to the reviewer for the time and effort devoted to our work and for the valuable observations and suggestions aimed at improving the paper.

Comment

Response

The authors have addressed the majority of my comments effectively, yet a few additional concerns persist. Firstly, the present study utilizes Machine Learning (ML) classification algorithms to predict parents’ school travel mode choices (i.e., motorized and non-motorized modes) based on responses to a diverse array of survey questions encompassing factors such as working hours, convenience, car ownership, and students’ safety. The title, “Assessing impact factors that affect school mobility utilizing a novel Artificial Intelligence approach,” needs revision as it fails to accurately convey the study’s purpose. Since using ML classification algorithms as an AI approach is a standard technique for prediction, as evidenced in Table 2, the term “novel Artificial Intelligence approach” should be adjusted to "Utilizing Machine Learning approach"

We revised the title of the paper accordingly.

Previous title: Assessing impact factors that affect school mobility utilizing a novel Artificial Intelligence approach

New title: Assessing impact factors that affect school mobility utilizing Machine Learning approach

Secondly, the authors mention that the processes of Exploratory Factor Analysis (EFA) and Confirmatory Factor Analysis (CFA) have been extensively detailed in prior research [1, 51, 52] to avoid unnecessary redundancy. However, the abstract suggests that "Survey questions are grouped into labeled factors using Exploratory and Confirmatory Factor Analysis." This implies that both EFA and CFA were indeed employed to identify and confirm factors in the present study. In light of this, it is imperative that the authors either present the results of the EFA and CFA in the current study or revise this line for clarity.

We revised the abstract for clarity, according to the reviewer's comment, as follows:

“The analysis and modeling of parameters influencing parents' decisions regarding school travel mode choice have perennially been a subject of interest. Concurrently, the evolution of Artificial Intelligence (AI) can effectively contribute to generating reliable predictions across various topics. This paper begins with a comprehensive literature review on classical models for predicting school travel mode choice, as well as the diverse applications of AI methods, with a particular focus on transportation. Building upon published questionnaire survey in the city of Thessaloniki (Greece) and the conducted analysis and exploration of factors shaping the parental framework for school travel mode choice, this study takes a step further: the authors evaluate and propose a Machine Learning (ML) classification model, utilizing the pre-recorded parental perceptions, beliefs, and attitudes as inputs to predict the choice between motorized or non-motorized school travel. The impact of potential changes in the input values of the ML classification model is also assessed. Therefore, the enhancement of sense of safety and security in the school route, the adoption of a more active lifestyle by parents, the widening of acceptance of public transportation, etc., are simulated, and the impact on the parental choice ratio between non-motorized and motorized school commuting is quantified”.

Given that new data from parents were collected, it is crucial to explicitly state whether CFA was conducted in the current study to confirm the factor structure for the gathered data. The inclusion of CFA results, if performed, will enhance the transparency of the analytical approach and provide readers with a comprehensive understanding of how factors were identified and validated in the specific context of the study.

We would like to clarify that there is no new data collection performed as part of the work presented in the current paper. The dataset we use in this paper was collected during previous works and is utilized here without any modifications or additions. Hence, there was no need to repeat any CFA or EFA, since they were already carried out and reported during previous studies, as referenced in the paper. CFA and EFA were outcomes from previous works and are only mentioned and referenced in this paper since their findings were some of the aspects that triggered the current research.

Therefore, we did not see any need or value in repeating and further emphasizing on aspects that have already been published in previous papers (1, 51, 52). Instead, we directly focus on the prospects and applicability of Machine Learning classifiers.

Reviewer 3 Report

Comments and Suggestions for Authors

The descriptive content of this article is inconsistent with the data in the table. It is recommended that the authors review the full text to avoid this situation. For example, Line 384-423 is inconsistent with the content in Table 3.

Author Response

First and foremost, we would like to express our warm thanks to the reviewer for the time and effort devoted to our work and for the valuable observations and suggestions aimed at improving the paper.

Comment

Response

The descriptive content of this article is inconsistent with the data in the table. It is recommended that the authors review the full text to avoid this situation. For example, Line 384-423 is inconsistent with the content in Table 3.

Apologies for the inconsistency. We revised the text accordingly to fix the problem.

Also, in lines 460-461, we added a short description for Table 4 to better clarify the use of the confusion matrix.

Round 3

Reviewer 3 Report

Comments and Suggestions for Authors

1.         This article drew a sample of 2747 students through questionnaires to serve as the source of data. However, there is no description of the population and whether the sample is sufficiently representative of the population.

2.         Line 253 indicated the gender ratio of the sample, but whether this gender distribution is consistent with the one of population. It was not mentioned in the article.

3.         Figure 3 shows the schematic content of the machine learning classifier. However, it does not explain the methods and mechanisms for applying machine learning.

4.         Table B1 presents the factor analysis results. However, what is the difference between confirmatory factor analysis and exploratory factor analysis in this article?

5.         Questionnaire Q15 has a total of 10 transport modes options. Why does this article divide the results into motorized transport mode and nonmotorized transport mode?

Author Response

3rd  Reviewer

First and foremost, we would like to express our warm thanks to the reviewer for the time and effort devoted to our work and for the valuable observations and suggestions aimed at improving the paper. 

This article drew a sample of 2747 students through questionnaires to serve as the source of data. However, there is no description of the population and whether the sample is sufficiently representative of the population.

The sample of 2747 students concerns a research presented as part of the literature review (lines 208-213, reference No. 49) and is not relevant to our research, which is briefly described in section 2.1. 

The questionnaire survey of the present research is described in lines 248-253 and pertained to a population of 100,000 students (as referred in line 253). The selected sample, as numerically outlined, ensured representativeness, as detailed in Eq. (1) and lines that follow (253-257).

Line 253 indicated the gender ratio of the sample, but whether this gender distribution is consistent with the one of population. It was not mentioned in the article.

Comment accepted.

In lines 258 to 267, authors revised manuscript, where additional information are given to validate the gender distribution consistency with the one of population. Specifically, relevant demographic data concerning Greece were added to transparently show the sample's representativeness.

Figure 3 shows the schematic content of the machine learning classifier. However, it does not explain the methods and mechanisms for applying machine learning.

Comment accepted.

We revised figure 3 to make it more informative, besides the fact that more details are in the text.

Still, figure 3 is kept simple for readability. All machine learning methods applied are referenced and explained in the text; please refer to lines 397 - 425. 

Table B1 presents the factor analysis results. However, what is the difference between confirmatory factor analysis and exploratory factor analysis in this article? 

Comment accepted. 

In the revised version of the manuscript, a concise description (and focus on the difference) of confirmatory factor analysis and exploratory factor analysis was added (lines 837-840). 

However, we kindly request the reviewer to consider that the factorial analyses, as stated in lines 82-87, 288-293, and in Appendix B, were previously performed in antecedent works by some of the authors of the current research, where the focus was on factorial analyses (EFA and CFA) and structural equation models describing school travel mode choice (References 1,52,53]. 

In the present study, the results from those previous publications were utilized to formulate classifiers predicting the school transport mode  and to investigate and quantify the influence of these factors on the changes in the percentages of motorized and non-motorized mobility. 

However, in order to facilitate the reader's understanding of which observed variables serve as inputs to the classifier and which of these observed variables constitute the latent variables, it was deemed advisable to include Appendix B.

A repetition of what has already been published would significantly increase the already expanded length of the work. Moreover, it would constitute a repetition of findings that have already been published.

Questionnaire Q15 has a total of 10 transport modes options. Why does this article divide the results into motorized transport mode and nonmotorized transport mode?

Comment accepted.

As illustrated in Figure 2, two modes of transport dominate school commuting in the study area: walking and using private motorized vehicles (car or school bus). The use of bicycles is limited (2%), so it can be integrated into non-motorized walking commuting. These results encourage a choice between non-motorized commuting (primarily walking) and motorized commuting. Since the study analyzes parents' preferences regarding their child's commuting, the authors decided to explore the impact of factors influencing parents to opt for a theoretically safer choice, such as motorized vehicles (private cars or buses), as opposed to a choice that incorporates the benefits of physical activity and environmental awareness, such as non-motorized transport.

Therefore, in the revised manuscript, lines 48-51 and 349-353 were added to emphasize the particular scientific interest and the utility-benefits of non-motorized school transport.
